# Two motor neuron synergies, invariant across ankle joint angles, activate the triceps surae during plantarflexion

Jackson Levine[1,2,3], Simon Avrillon[1,2,4] 🆔, Dario Farina[4] 🆔, François Hug[5,6] 🆔 and José L. Pons[1,2,3]

[1]*Legs + Walking Lab, Shirley Ryan AbilityLab, Chicago, IL, USA*

[2]*Department of Physical Medicine and Rehabilitation, Feinberg School of Medicine, Northwestern University, Chicago, IL, USA*

[3]*Department of Biomedical Engineering, McCormick School of Engineering, Northwestern University, Chicago, IL, USA*

[4]*Department of Bioengineering, Faculty of Engineering, Imperial College London, London, UK*

[5]*Université Côte d'Azur, LAMHESS, Nice, France*

[6]*School of Biomedical Sciences, The University of Queensland, St Lucia, Queensland, Australia*

Handling Editors: Richard Carson & Madeleine Lowery

The peer review history is available in the Supporting Information section of this article (https://doi.org/10.1113/JP284503#support-information-section).

*The Journal of Physiology*

**Abstract** Recent studies have suggested that the nervous system generates movements by controlling groups of motor neurons (synergies) that do not always align with muscle anatomy. In this study, we determined whether these synergies are robust across tasks with different mechanical constraints. We identified motor neuron synergies using principal component analysis (PCA) and cross-correlations between smoothed discharge rates of motor neurons. In part 1, we used

J. Levine and S. Avrillon contributed equally to this work.

This article was first published as a preprint. Levine J, Avrillon S, Farina D, Hug F, Pons JL. 2022. Two motor neuron synergies, invariant across ankle joint angles, activate the triceps surae during plantarflexion. bioRxiv. https://doi.org/10.1101/2022.11.11.516183

The Journal of Physiology

simulations to validate these methods. The results suggested that PCA can accurately identify the number of common inputs and their distribution across active motor neurons. Moreover, the results confirmed that cross-correlation can separate pairs of motor neurons that receive common inputs from those that do not receive common inputs. In part 2, 16 individuals performed plantarflexion at three ankle angles while we recorded EMG signals from the gastrocnemius lateralis (GL) and medialis (GM) and the soleus (SOL) with grids of surface electrodes. The PCA revealed two motor neuron synergies. These motor neuron synergies were relatively stable, with no significant differences in the distribution of motor neuron weights across ankle angles ($P = 0.62$). When the cross-correlation was calculated for pairs of motor units tracked across ankle angles, we observed that only 13.0% of pairs of motor units from GL and GM exhibited significant correlations of their smoothed discharge rates across angles, confirming the low level of common inputs between these muscles. Overall, these results highlight the modularity of movement control at the motor neuron level, suggesting a sensible reduction of computational resources for movement control.

(Received 5 February 2023; accepted after revision 10 August 2023; first published online 24 August 2023)

**Corresponding author** S. Avrillon: Department of Bioengineering, Imperial College London, 86 Wood Ln, London W12 0BZ, UK. Email: s.avrillon@imperial.ac.uk

J.L. Pons: Legs + Walking Lab, Shirley Ryan AbilityLab, Floor 23, 355 E Erie St, Chicago, IL 60611, USA. Email: jpons@sralab.org

**Abstract figure legend** Here, we aimed to test the robustness of motor neuron synergies across isometric plantarflexion tasks performed at different ankle angles. We found with simulations that our methods, i.e. principal component analysis and correlation of smoothed discharge rates, can accurately identify the number of common inputs and whether pairs of motor neurons receive or do not receive a significant proportion of common inputs. Thus, we identified two main common inputs driving motor neurons innervating the gastrocnemius lateralis (GL), gastrocnemius medialis (GM) and soleus (SOL), with no significant systematic changes in the distribution of weights across ankle angles. We also found that a low percentage of pairs of motor units from GL and GM received a significant proportion of common inputs.

## Key points

- The CNS might generate movements by activating groups of motor neurons (synergies) with common inputs.
- We show here that two main sources of common inputs drive the motor neurons innervating the triceps surae muscles during isometric ankle plantarflexions.
- We report that the distribution of these common inputs is globally invariant despite changing the mechanical constraints of the tasks, i.e. the ankle angle.
- These results suggest the functional relevance of the modular organization of the CNS to control movements.

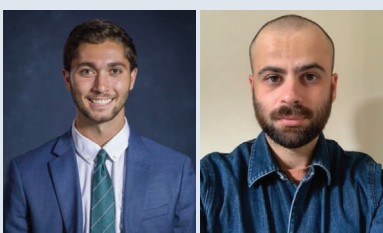

**Jackson Levine** is a PhD candidate in the biomedical engineering department at Northwestern University conducting research at the Shirley Ryan AbilityLab. He received a BSE in Biomedical Engineering and a BS in Neuroscience from Tulane University. His current work focuses on assessment of muscle activity to investigate neural coordination strategies in healthy individuals and individuals post-stroke, with the goal of developing improved biomarkers and rehabilitation strategies. **Simon Avrillon** is a research associate at the Department of Bioengineering, Imperial College London. He received his PhD in human movement sciences from Paris-Saclay University in 2019 and completed his first postdoctoral position from 2019 to 2021 at the Shirley Ryan AbilityLab, Chicago. His research currently focuses on motor control in healthy people and patients through the recording of motor neuron firing activity identified from EMG signals.

## Introduction

The neural control of movement relies on the coordinated activity of numerous motor units. A large proportion of these motor units receives common synaptic inputs (Laine et al., 2015; Negro et al., 2016b), which arise from various sources, including descending commands and proprioceptive feedback (Glover & Baker, 2022; Heckman & Binder, 1993; Heckman & Enoka, 2012; Laine et al., 2014; Lemon, 2008). Simulation studies have shown that the pool of motor neurons tends to transmit these common synaptic inputs linearly to the muscle while it attenuates independent inputs (Farina et al., 2014b). Thus, the time-varying modulation of muscle force during a sustained isometric contraction depends mainly on the modulation of common synaptic inputs, as evidenced by the significant correlation between the low-pass-filtered cumulative spike train of motor units and muscle force (De Luca & Erim, 2002; Negro et al., 2009; Thompson et al., 2018).

When considering the coordination of multiple muscles, common synaptic inputs can also be distributed across groups of motor units that share a similar function to generate a force in a desired direction (Bizzi & Cheung, 2013; Tresch & Jarc, 2009). This distribution outlines a modular organization of low dimensionality, theorized as synergies or motor modules and demonstrated mainly at the muscle level by combining the recording of multiple muscles with bipolar EMG and factorization algorithms (d'Avella et al., 2003; Ting et al., 2015; Tresch et al., 1999; Yaron et al., 2020). The recent development of algorithms that discriminate the discharge activity of individual motor units has changed the scale of this approach, with synergies now being considered at the motor neuron level. These motor neuron synergies can group subsets of motor neurons within each muscle or across portions of different muscles (Hug et al., 2023b). For example, it has been shown that motor neurons innervating a single muscle can be separated into several synergies based on their fluctuations of discharge rate, even during tasks of a single degree of freedom (Del Vecchio et al., 2023; Madarshahian et al., 2021).

To date, the identification of motor neuron synergies has been reported only for motor tasks with stable mechanical constraints (Del Vecchio et al., 2023; Hug et al., 2023b; Madarshahian et al., 2021). It remains unknown whether the distribution of common inputs to groups of motor neurons is robust across motor tasks, as observed for muscle synergies (d'Avella et al., 2003). Addressing this question is fundamental to prove that the observed motor neuron synergies feature an effective way of simplifying the neural control of movement through motor modules that are invariant across a repertoire of tasks.

The technical challenge of identifying motor neuron synergies across tasks is that motor neurons must be tracked across contractions. However, EMG decomposition is currently applicable only during isometric contractions with slight changes in muscle length (Farina & Holobar, 2016). Indeed, this procedure relies on the stability of muscle geometry to identify consistent spatiotemporal profiles of motor unit action potentials with the grid of surface electrodes (Del Vecchio & Farina, 2019; Martinez-Valdes et al., 2017; Oliveira & Negro, 2021). For this reason, we identified in this study motor neurons innervating the triceps surae muscles [soleus (SOL), gastrocnemius medialis (GM) and gastrocnemius lateralis (GL)] and tracked them during isometric contractions performed at three different ankle angles. These muscles are interesting because an uncoupling of their neural drives has been demonstrated during sustained contractions (Hug et al., 2021b; Rossato et al., 2022), despite all being connected to the same distal tendon and being grouped within the same muscle synergy during most natural behaviours. Likewise, it was shown that changes in neural drive, owing to the alteration of force-generating capacity during an ankle rotation, might differ between GL and GM/SOL (Lacourpaille et al., 2017). This suggests that the CNS might control several motor modules, even during a task with a single degree of freedom (Lacourpaille et al., 2017; Rossato et al., 2022).

In this study, we identified motor neuron synergies by estimating the distribution of common synaptic inputs to groups of motor neurons innervating the triceps surae muscles. To this end, we used principal component analysis (PCA) and cross-correlation of their smoothed discharge rates (Fig. 1*A*). To validate our approach, we initially simulated 100 motor neurons receiving two sources of common synaptic input, and we assessed whether the proposed methods could identify the low-dimensional control of motor neuron behaviour. Then, we recorded EMG signals from the triceps surae muscles in 16 participants to evaluate the robustness of motor neuron synergies across tasks. We hypothesized that: (i) despite the single dimension captured by the classic EMG analysis at the muscle level (e.g. van den Hoorn et al., 2015), the dimensionality of control would be greater than one at the motor neuron level; and (ii) the structure of the motor neuron synergies would remain invariant across joint angles. The validation of these hypotheses would prove that the CNS controls movements by distributing common inputs to stable groups of motor neurons, ensuring a sensible reduction of dimensions for movement control.

## Methods

### Part 1: simulations

Initially, we aimed to test the sensitivity of our methods to capture the low-dimensional control of motor neurons (Fig. 1*B*). Given that previous studies have shown different

changes in neural drive between GL and GM/SOL during constrained isometric plantarflexions (Hug et al., 2021b; Lacourpaille et al., 2017), we hypothesized that two main sources of common inputs might drive the triceps surae muscles during plantarflexion. Thus, we simulated 100 motor neurons receiving common and independent synaptic inputs with the model developed by Elias & Kohn (2013), with one compartment for the soma and one compartment for the dendritic tree. This model is an updated version of a previously published motor neuron model (Cisi & Kohn, 2008), with the addition of a L-type $Ca^{2+}$ channel into the dendritic compartment to consider the non-linearity between motor neuron inputs and outputs attributable to persistent inward currents (Heckman & Enoka, 2012). Geometric and electronic parameters of the 100 motor neurons were linearly interpolated from the range of values reported by Elias & Kohn (2013; their table 1) for an S-type motor unit.

The total synaptic input for each motor neuron was the linear summation of common and independent inputs simulated as white Gaussian noise. The common and independent inputs had equal variances, with a bandwidth of 0–2.5 Hz for the common inputs and a bandwidth of 0–50 Hz for the independent inputs. The mean and the variance of the total synaptic input were tuned to generate physiologically realistic distributions of interspike intervals (Negro et al., 2016b). To simulate motor neuron synergies, we generated two uncorrelated sources of common synaptic inputs with a range of synaptic weights for the pool of motor neurons. Specifically, 20 motor neurons received 100% of common input (CI) #1, 20 motor neurons received 100% of CI#2, and 60 motor neurons received CI#1 and CI#2 with synaptic weights ranging from 99% CI#1/1% CI#2 to 1% CI#1/99% CI#2. The synaptic weights were shuffled across the 100 motor neurons and fixed for all the simulations.

*A*  Schematic representation of the framework to identify MN synergies

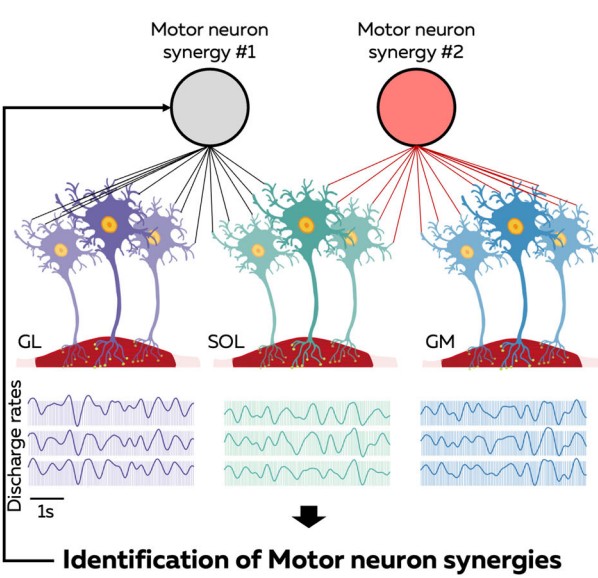

*B*  Overview of the simulation protocol

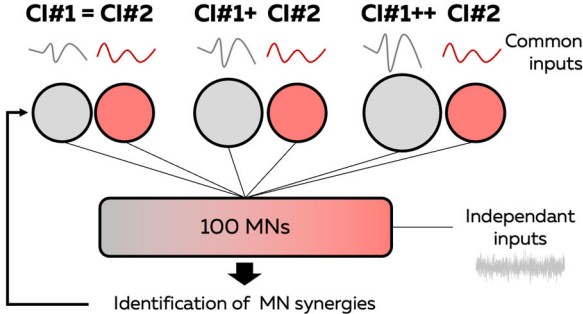

*C*  Overview of the experimental protocol

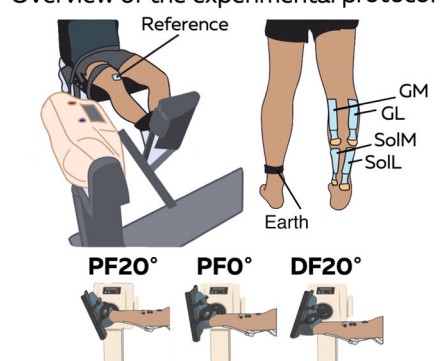

**Figure 1. Overview of the study**

We aimed to identify motor neuron synergies driving the triceps surae muscles during ankle plantarflexion. *A*, we hypothesized that two motor neuron synergies would drive the gastrocnemius lateralis (GL), soleus (SOL) and gastrocnemius medialis (GM) muscles. To test this hypothesis, we needed to identify the distribution of common inputs to the pool of motor neurons. Thus, we applied principal component analysis and cross-correlation on motor neuron smoothed discharge rates. *B*, initially we assessed the ability of these methods to identify the distribution of common inputs across motor neurons using 100 simulated motor neurons driven by two sources of common inputs with known synaptic weights. We changed the amplitude of these inputs to mimic the divergent variations in discharge rate observed experimentally between GL and GM/SOL muscles. *C*, then, we identified and tracked motor units from EMG signals recorded from the GL, GM and SOL muscles across three ankle angles. As for the simulation, we determined whether the distribution of common inputs to the tracked motor units was robust across the three conditions using principal component analysis and cross-correlation. Abbreviations: CI, common input; PF, plantarflexed; MN, motor neuron. [Colour figure can be viewed at wileyonlinelibrary.com]

We generated three simulations with different amplitudes of the two common inputs. In the first simulation, the amplitudes of common inputs #1 and #2 were equal (CI#1 = CI#2). In the second (CI#1+ CI#2) and third simulations (CI#1++ CI#2), the amplitude of common input #1 was amplified to generate an averaged increase in discharge rate of 1 pulse per second (pps) between each condition for all the motor neurons receiving this input (Fig. 1*B*). This idea was to mimic the divergent modulation of two separate sources of common synaptic inputs and to match the discharge rates observed experimentally during the plantarflexions.

## Part 2: experiments

In the second part of the study, we identified and tracked the motor neuron synergies driving the triceps surae muscles during plantarflexions performed at three ankle joint angles (Fig. 1*C*).

**Ethical approval and participants.** A sample of 16 individuals, with an equal number of males ($n = 8$; age, $28 \pm 4$ years; height, $178 \pm 6$ cm; body mass, $77 \pm 10$ kg) and females ($n = 8$; age, $29 \pm 11$ years; height, $162 \pm 8$ cm; body mass, $60 \pm 9$ kg), volunteered to participate in this experiment. They had no history of knee/ankle injury in the last 6 months before the experiments nor lower leg pain that would impact their ability to complete the voluntary contractions. The institutional review board of Northwestern University reviewed and approved the protocol of this study (STU00212191), which followed the standards of the *Declaration of Helsinki*. Participants provided their informed written consent before starting the experimental session.

**Protocol.** Participants sat on a dynamometer (Biodex System 4 Pro; Biodex Medical, Shirley, NY, USA) with their hip flexed at 90° (0° being the neutral position) and their knee fully extended (Fig. 1*C*). The foot of the dominant leg (i.e. right leg for all the participants) was fixed to a pedal that held the ankle at specified angles (Fig. 1*C*). Two inextensible straps immobilized their torso, while additional straps around the waist and the thigh of the dominant leg ensured that force generation relied solely on the calf muscles.

All experimental tasks consisted of isometric ankle plantarflexion. Initially, the participants performed a standardized warm-up, with their ankle positioned in the neutral position (0°, i.e. the foot is perpendicular to the shank). Participants were asked to push against the pedal with gradually increasing effort, beginning at ∼50% of their subjective maximum capabilities. After three contractions, participants were asked to increase effort until they reached ∼80–90% of their subjective maximum capabilities.

After the warm-up, experimentation began. The experimental protocol was repeated for three different conditions in a randomized order: the ankle plantarflexed at 20° (PF20°), at 0° (PF0°) and dorsiflexed at 20° (DF20°) (Fig. 1*C*). For each condition, participants performed two maximal isometric voluntary contractions (MVCs) for 3–5 s, with 60 s of rest in between. The peak torque was calculated using a moving average window of 250 ms, and the maximal value was used to set the target of the sub-maximal isometric contractions at 20% of the MVC. After 60 s of rest, participants performed three trapezoidal contractions, consisting of a 15 s linear ramp-up, 30 s plateau at 20% of the MVC, and 15 s linear ramp-down. These three trapezoidal contractions were separated by 10 s of rest. Target and torque output feedback were displayed on a monitor in front of the participants. The EMG acquisition system (EMG-Quattrocento, 400-channel EMG amplifier; OT Bioelettronica, Italy) was used to digitize the torque signal at a sampling rate of 2048 Hz. Torque signals were low-pass filtered (third order Butterworth filter, with a cut-off frequency of 20 Hz) and baseline corrected to remove the weight of the leg and the pedal. The torque accuracy was calculated as the root mean squared error (RMSE) of the signal over the plateau.

**EMG recording.** Surface EMG signals were recorded from the GM, the GL and the lateral and medial regions of the SOL using two-dimensional adhesive grids of 64 electrodes (GR08MM1305, $13 \times 5$ gold-coated electrodes with one electrode absent on a corner; interelectrode distance, 8 mm; OT Bioelettronica, Italy). The grids were aligned in the fascicle direction for the GM and GL. The grids over the soleus were positioned on each side of the Achilles tendon in the muscle shortening direction. Before placing the electrodes, the skin was shaved and cleaned with an abrasive pad and an alcohol wipe. A semi-disposable bi-adhesive foam layer was used to hold the grids of electrodes to the skin, and conductive paste filled the cavities of the adhesive layers to make skin–electrode contact. Reference electrodes (Ag/AgCl snap electrodes; Noraxon, USA) were placed over the tibia. The EMG signals were recorded in monopolar mode, bandpass filtered at 10–500 Hz, and digitized at a sampling rate of 2048 Hz using the multichannel EMG acquisition system (EMG-Quattrocento, 400 channel EMG amplifier; OT Bioelettronica, Italy).

## Data analysis

Data were analysed using MATLAB custom-written scripts (R2021a; The MathWorks, Natick, MA, USA).

**EMG decomposition.** The monopolar EMG signals were bandpass filtered between 20 and 500 Hz with a second-order Butterworth filter. After visual inspection, channels with low signal-to-noise ratio or artefacts were discarded. The EMG signals were then decomposed into motor unit pulse trains with a blind-source separation algorithm (Negro et al., 2016a). In short, the EMG signals were first extended and whitened. Thereafter, a fixed-point algorithm optimized filters that maximize the level of sparsity of the identified sources of the EMG signal, i.e. the motor unit pulse trains. The high peaks of the motor unit pulse trains were considered as discharge times and separated from the noise using *K*-means classification. A second algorithm refined the estimation of the discharge times by minimizing the coefficient of variation of the interspike intervals while iteratively optimizing motor unit filters. This decomposition procedure has been validated using experimental and simulated signals (Negro et al., 2016a). After the automatic identification of the motor units, all the motor unit pulse trains were inspected visually. The automatic detection of discharge times was edited manually for false positives and false negatives when necessary (Del Vecchio et al., 2020; Hug et al., 2021a). Motor units with a pulse-to-noise ratio of >30 dB after manual editing were retained for further analysis (Holobar et al., 2014).

**Crosstalk and duplicates between motor units.** We checked all identified motor units carefully for potential crosstalk or duplicates between the two SOL locations, thus ensuring the uniqueness of each motor unit within the dataset (Hug et al., 2021b). To this end, we used the discharge times as triggers to segment the EMG signals differentiated in the column direction for each grid and identify the profiles of the motor unit action potentials. Then, we calculated the average peak-to-peak amplitude of the action potentials for all grids. We considered that a motor unit was falsely associated with a muscle owing to crosstalk when the peak-to-peak amplitude was larger in a neighbouring muscle than in the muscle of origin. On average, 2.6 ± 2.5 motor units were discarded from the analysis per participant. In addition, we verified that no motor units were identified from both grids located over the lateral and the medial parts of the SOL. We considered motor units as duplicates when they shared ≥30% of their discharge times with a tolerance of one data point (i.e. 0.5 ms; Holobar et al., 2014). At the end of these analyses, motor units identified over the lateral and medial parts of the SOL were pooled, with on average 0.7 ± 1.4 motor units discarded per participant.

**Motor unit tracking.** To assess whether a change in muscle length impacts the distribution of common synaptic inputs and motor unit behaviour, we matched motor units across ankle angles. For this, we used the unique spatiotemporal profiles of motor unit action potentials within the grids of electrodes (Del Vecchio & Farina, 2019; Martinez-Valdes et al., 2017; Oliveira & Negro, 2021). EMG signals were first differentiated in the column direction to obtain 59 single-differential EMG signals. We then identified motor unit action potential profiles by spike-triggered averaging each of the 59 EMG signals. Given that the position of the action potentials within the window might differ between motor units, we considered the local maxima, i.e. the positive peak of the action potential with the highest energy within the grid, as the centre of the 50 ms window. Finally, we concatenated the motor unit action potentials from the 59 channels, and we performed a two-dimensional cross-correlation between motor units. Pairs with a correlation coefficient of >0.75 were considered as matches (Del Vecchio & Farina, 2019; Martinez-Valdes et al., 2017). The matches were displayed and inspected visually to guarantee the similarity of motor unit action potentials. Motor units were matched separately between PF20° and PF0° and between DF20° and PF0°. Only motor units matched across the three conditions were considered for the analyses. The average discharge rate was estimated over the torque plateau for all tracked motor units.

**Motor neuron synergies.** We identified the common dynamics of motor unit smoothed discharge rates in parts 1 and 2 using PCA. The motivation behind this analysis was initially to assess the ability of PCA to identify the number of common inputs accurately, and then to estimate the relative distribution of these common inputs or motor neuron synergies across motor neurons. The discharge activities of all motor neurons were converted into binary vectors with discharge times equal to one. These binary vectors were then convoluted with a 400 ms Hanning window, such that the correlation was calculated from the oscillations of the discharge rate related to the fluctuation of force (De Luca & Erim, 2002). It is noteworthy that focusing on the lowest bandwidth also limits the effect of the non-linear relationship between the synaptic input and the output signal (Negro & Farina, 2012). These signals were high-pass filtered to remove offsets and trends using a second order Butterworth filter with a cut-off frequency of 0.75 Hz. Finally, we normalized the smoothed discharge rates between zero and one to avoid any bias of the PCA output related to differences in discharge rates between motor neurons and ankle angles.

Normalized smoothed discharge rates were then concatenated in a $n \times T$ matrix, where $n$ is the number of matched motor neurons and $T$ the number of time samples. We applied PCA to this matrix and identified $n$ components ranked with the level of variance explained by each component. To select the number of components of interest (i.e. motor neuron synergies), we fitted portions of the curve of the variance explained with straight lines

by iteratively decreasing the number of points considered in the analysis. The number of components of interest was considered as the first data point where the mean squared error of the fit falls under $5e^{-4}$ (Cheung et al., 2005; d'Avella et al., 2011). In our study, this threshold was reached with two components for both the simulated and experimental data. Then, we displayed two-dimensional biplots, where $x$ and $y$ coordinates were the weights of each motor neuron in the two first components, and we calculated the angle between vectors. We considered that the higher the angle between two vectors, i.e. two motor neurons, the lower the proportion of inputs shared between these motor neurons. Therefore, we described the changes in the distribution of common synaptic inputs to motor neurons by comparing the angles between conditions.

**Pairs of motor neurons receiving common synaptic inputs.**
For both part 1 and part 2, we estimated, for each pair of motor neurons, whether they received a significant level of common synaptic input. To this end, we calculated the cross-correlation between their smoothed discharge rates over a 10 s window that maximized the number of motor neurons continuously firing during the plateaus of the sub-maximal contractions. The presence of common synaptic input for each pair of motor neurons was estimated through the maximal cross-correlation coefficient with a maximal lag of ±100 ms (De Luca & Erim, 2002). We considered that a pair of motor neurons received common synaptic input when the cross-correlation coefficient was statistically significant (Rodriguez-Falces et al., 2017). Specifically, we defined a significance threshold as the 99th percentile of the cross-correlation coefficient distribution generated with resampled versions of all series of discharge times per participant and condition. To this end, we generated random series of discharge times for each motor neuron by bootstrapping the interspike intervals (random sampling with replacement). This random series of discharge times had the same number of discharge times and the same discharge rate (mean and SD) as the original series of discharge times. We repeated this step four times per pair of motor neurons.

## Statistics

All statistical analyses were performed with RStudio (USA). Initially, quantile–quantile plots and histograms were displayed to check the normality of the data distribution. If the data were determined not to be normal, they were transformed to remove the skew. All statistical analyses were performed using linear mixed effect models implemented in the R package *lmerTest* with the Kenward–Roger method to estimate the denominator degrees of freedom and the *P*-values. This method considers the dependence of data points

(i.e. individual motor neurons) within each participant owing to, e.g. common synaptic inputs. When necessary, multiple comparisons were performed using the R package *emmeans*, which adjusts the *P*-value using Tukey's method. The significance level was set at 0.05. Values are reported as the mean ± SD.

In part 1, we assessed the ability of our methods to identify the number of synergies and classify the motor neurons according to the weights of the common inputs they receive. After verifying that the PCA predicts the number of motor neuron synergies accurately, we assessed the correlation between the angles calculated between vectors from the biplots and the level of common inputs. Then, we measured the level of common inputs between motor neurons having significant and non-significant cross-correlation coefficients between their smoothed discharge rates across simulations. In this way, we aimed to quantify the minimal level of common input needed to reach the level of significance in the three conditions and assess the maximal level of common input, if any, needed to have non-significant correlations in the three conditions.

In part 2, we initially compared MVC and torque RMSE using a linear mixed effects model, with condition (PF20°, PF0° or DF20°) as a fixed effect and participant as a random effect. The averaged discharge rates of matched motor units were compared using a linear mixed effects model, with condition (PF20°, PF0° or DF20°) and muscle (GL, GM or SOL) as fixed effects and participant as a random effect.

Then, we used PCA to report the number of motor neuron synergies at each ankle angle for each participant. We calculated the angles between all the motor neurons, and we used a linear mixed effects model to compare these angles, with muscle (GL, GM, SOL, GL–GM, GL–SOL or GM–SOL) and condition (PF20°, PF0° or DF20°) as fixed effects and participant as a random effect. Finally, we assessed the level of common synaptic input to pairs of motor neurons from the same and different muscles using the cross-correlation coefficients between smoothed discharge rates. We used a linear mixed effects model to compare correlation coefficients, with muscle (GL, GM, SOL, GL–GM, GL–SOL or GM–SOL) and condition (PF20°, PF0° or DF20°) as fixed effects and participant as a random effect. The ratio of pairs of motor neurons with correlation coefficients above the significance threshold at each ankle angle was reported, as was the ratio of pairs of motor neurons with significant or non-significant correlations across all angles.

## Results

### Part 1: simulations

Initially, we investigated the ability of our approach to capture the low-dimensional control of motor neurons

with PCA (Fig. 2*A-C*). The cumulative sum of the variance explained reached a plateau with two components for all simulations (Fig. 2*D*). The value of variance explained with two components was 68.8, 70.9 and 72.1% for conditions CI#1 = CI#2, CI#1+ CI#2 and CI#1++ CI#2, respectively. The angles between motor neurons ranged between 0 and 135° across the three conditions (Fig. 2*F*). We computed correlations between the angles and the difference in ratio of common inputs, calculated as the differences in weights of CI#1 between the two motor neurons. For example, two motor neurons receiving 100% CI#1/0% CI#2 and 0% CI#1/100% CI#2 have a difference in ratio equal to one. Adjusted $R^2$ values reached 0.91, 0.91 and 0.92 for conditions CI#1 = CI#2, CI#1+ CI#2 and CI#1++ CI#2, respectively. This demonstrates that

the angle between two motor neurons in the biplots is explained largely by the ratio of common inputs they receive, i.e. the greater the angle, the higher the difference in ratio of common inputs (Fig. 2*F*).

We then performed cross-correlation between smoothed discharge rates of individual motor neurons with a significance threshold fixed at 0.29 (Fig. 2*E*). Correlation coefficients are displayed in Fig. 2*E* according to the difference in ratio of common inputs. Motor neurons with a difference in ratio of common inputs of 0.29 ± 0.23, 0.30 ± 0.23 and 0.30 ± 0.23 had a significant correlation between their smoothed discharge rates in conditions CI#1 = CI#2, CI#1+ CI#2 and CI#1++ CI#2, respectively. Conversely, motor neurons with a difference in ratio of common inputs of 0.88 ± 0.13

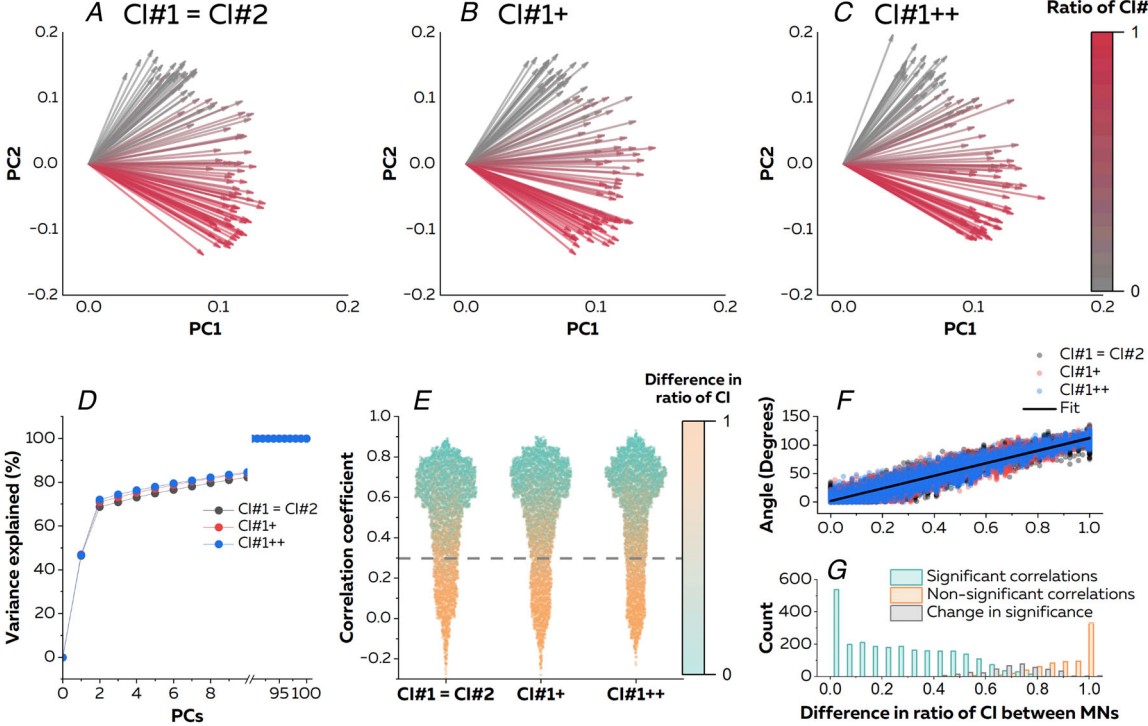

**Figure 2. Simulation of a pool of motor neurons with two sources of common synaptic inputs**
We simulated 100 motor neurons (MNs) using an open-source model (Elias & Kohn, 2013). Each motor neuron received 100% common input #1 (CI#1, 20 motor neurons), 100% common input #2 (CI#2, 20 motor neurons) or a ratio of CI#1 and CI#2 ranging from 99% CI#1/1% CI#2 to 1% CI#1/99% CI#2 (60 motor neurons). We generated three simulations with different amplitudes of CI#1 to mimic the divergent modulation of two separate sources of common synaptic inputs and to match the discharge rates experimentally observed during the plantarflexions. *A–C*, we applied principal component analysis and displayed the two-dimensional biplots based on the two main components (PC1 and PC2). *D*, the cumulative sum of the variance explained reached a plateau with two components. Each vector represents an individual motor neuron, with the colour depending on the ratio of CI#1 received by the motor neuron. *E*, next, we estimated the level of common synaptic input for all pairs of motor neurons using cross-correlation analyses between individual motor neuron smoothed discharge rates. All correlations >0.29 (dashed grey line) were significant. The colour of scatters depends on the difference in the ratio of CI#1 between motor neurons in each pair. For example, two motor neurons receiving 100% CI#1/0% CI#2 and 0% CI#1/100% CI#2 have a difference in ratio equal to one. *F*, angles between motor neurons were explained largely by differences in the ratio of CI#1. *G*, when we modulated the amplitude of CI#1 and CI#2 across simulated conditions, a portion (albeit small) of the pairs exhibited variable levels of correlation. The histogram is a distribution of the difference in ratio for CI#1 for each pair of motor neurons with significant, non-significant or a different level of significance across simulations. [Colour figure can be viewed at wileyonlinelibrary.com]

had non-significant correlation between their smoothed discharge rates, regardless of the condition (Fig. 2*E*). We found that 68.5% of the pairs of motor neurons exhibited significant correlations between their smoothed discharge rates across the three simulations, whereas only 20.1% of the pairs exhibited non-significant correlations. These pairs of motor neurons with consistent non-significant correlations across simulations mostly received different inputs, with a difference in ratio of common inputs of 0.93 ± 0.09 (Fig. 2*G*). Pairs of motor neurons with consistent significant correlations across simulations mostly received the same inputs, with a difference in ratio of common inputs of 0.27 ± 0.21 (Fig. 2*G*). We found that 11.4% of the pairs of motor neurons had changes in significance across simulated conditions, with differences in the ratio of common inputs of 0.71 ± 0.12 (Fig. 2*G*). This showed that changes in the level of correlation in the cross-correlation analysis can be explained by the amplification of one of the common synaptic inputs, despite fixed ratios of common inputs between simulations. It is noteworthy that this affected a minor proportion of pairs of motor neurons (11.5%) and that motor neurons receiving low or high proportions of the same common inputs had the same level of correlation between the simulations.

## Part 2: experiments

**Torque.** As expected, the MVC torque was significantly different between conditions ($F = 36.94$, $P < 0.001$), with MVC torque at PF20° (383.4 ± 225.4 N m) being significantly lower than that at PF0° (715.7 ± 354.8 N m, $P < 0.001$) and DF20° (790.0 ± 448.2 N m, $P < 0.001$). There was no difference between PF0° and DF20° ($P = 0.892$). The RMSE was not significantly different between conditions ($F = 2.73$, $P = 0.082$), meaning that the target torque was matched with similar accuracy across conditions.

**Motor unit discharge characteristics.** A total of 595 (37 ± 22 per participant; range, 5–79), 613 (38 ± 27 per participant; range, 3–96) and 471 (29 ± 22 per participant; range, 3–76) motor units were identified at PF20°, PF0° and DF20°, respectively. They had an average pulse-to-noise ratio of 33.3 ± 2.6 dB. Ninety-six motor units (17.2%) were matched over the three conditions, with 17, 45 and 34 motor units for the GL, GM and SOL, respectively. It is noteworthy that six participants were excluded from the analyses, because no motor units could be matched across conditions, and thus data in this section are reported for the 10 remaining participants. We first compared the discharge characteristics of these tracked motor units (Fig. 3). When considering the average discharge rates, significant effects of condition ($F = 18.89$, $P < 0.001$), muscle ($F = 3.41$, $P = 0.034$) and an interaction condition × muscle ($F = 2.56$, $P = 0.039$) were observed. The average discharge rate of motor units from the GL was significantly lower at DF20° (8.1 ± 1.8 pps) than at PF20° (10.5 ± 2.5 pps). Likewise, the average discharge rate of motor units from the SOL was significantly lower at DF20° (7.8 ± 2.3 pps) than at PF20° (9.1 ± 2.7 pps). No significant differences were observed for other angles or muscles (all $P > 0.109$).

**Number of common inputs.** The motor neuron synergy analysis was performed for participants with motor units identified from all muscles in the three conditions ($n = 12$ participants; 34 ± 17 motor neurons per participant). Principal component analysis was performed on the normalized smoothed discharge rates. Initially, we

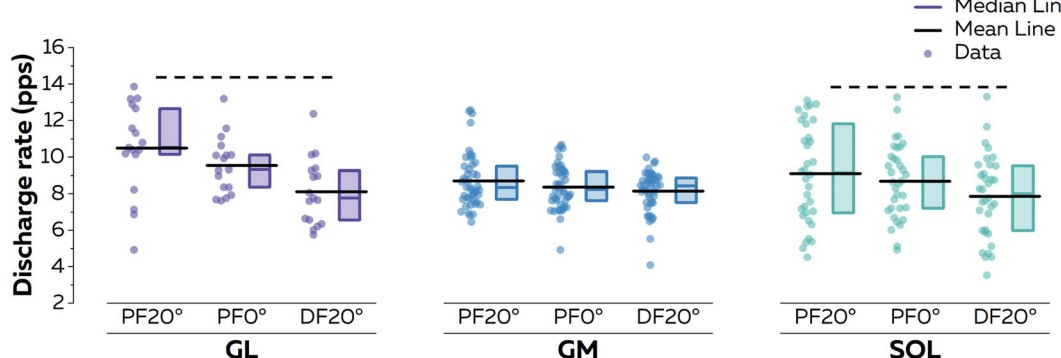

**Figure 3. Motor unit discharge rate**
We compared the discharge rate of all matched motor units. The average discharge rate was estimated after smoothing instantaneous discharge rates with a 1 s Hanning window over the plateau. Each scatter represents an individual motor unit. The horizontal dashed lines above the plots depict the significant statistical differences with $P < 0.05$. For the sake of clarity, we report only the differences within muscles. Abbreviations: DF, dorsiflexion; GL, gastrocnemius lateralis; GM, gastrocnemius medialis; PF, plantarflexion; and SOL, soleus. [Colour figure can be viewed at wileyonlinelibrary.com]

intended to identify the number of common inputs or motor neuron synergies. To this end, we used mean square error to determine the number of components where the cumulative sum of the variance explained reached a plateau (Cheung et al., 2005; d'Avella et al., 2011). We found, on average, two motor neuron synergies for the three conditions. The two-dimensional biplots, with the weight of each motor neuron in the two motor neuron synergies, are displayed in Fig. 4.

We then calculated the angles for each pair of vectors (i.e. each pair of motor neurons, $n = 26{,}589$; Fig. 5*A*) in the biplot. Simulations demonstrated that these angles provide a good estimate of the difference in the ratio of common inputs between two motor neurons (Fig. 2*F*). For each of the 12 participants, a mean angle was calculated for each muscle and condition (Fig. 5*B*). We found a significant main effect of muscle ($F = 22.28$; $P < 0.001$), but neither a significant main effect of condition ($F = 0.47$; $P = 0.62$) nor a significant interaction ($F = 1.30$; $P = 0.23$). Specifically, angles were significantly lower between motor

neurons within the GL ($33.0 \pm 22.3°$; $P = 0.005$) and within the GM ($23.6 \pm 18.2°$; $P < 0.001$) than between pairs of motor units within the SOL ($50.7 \pm 26.6°$; Fig. 5*B*). Moreover, the angles between motor neurons within the GL and within the GM were also significantly lower than the angles between motor neurons from different muscles, i.e. GL–GM, GL–SOL and GM–SOL (all $P < 0.001$; Fig. 5*B*). Finally, the angles between motor neurons within the SOL were lower than the angles between motor units of GM–SOL ($68.7 \pm 15.9°$; $P = 0.006$). The differences between pairs of muscles are displayed in Fig. 5. Taken together, these results suggest that motor units within the GL or within the GM receive the highest proportion of common input.

**Distribution of common inputs across motor neurons.** We analysed the correlation for discharge rates of 416 pairs of matched motor units (Fig. 6*A*). Specifically, we had 34, 147 and 92 pairs of motor neurons for GL, GM and SOL, respectively, and 46, 29 and 68 pairs of motor neurons

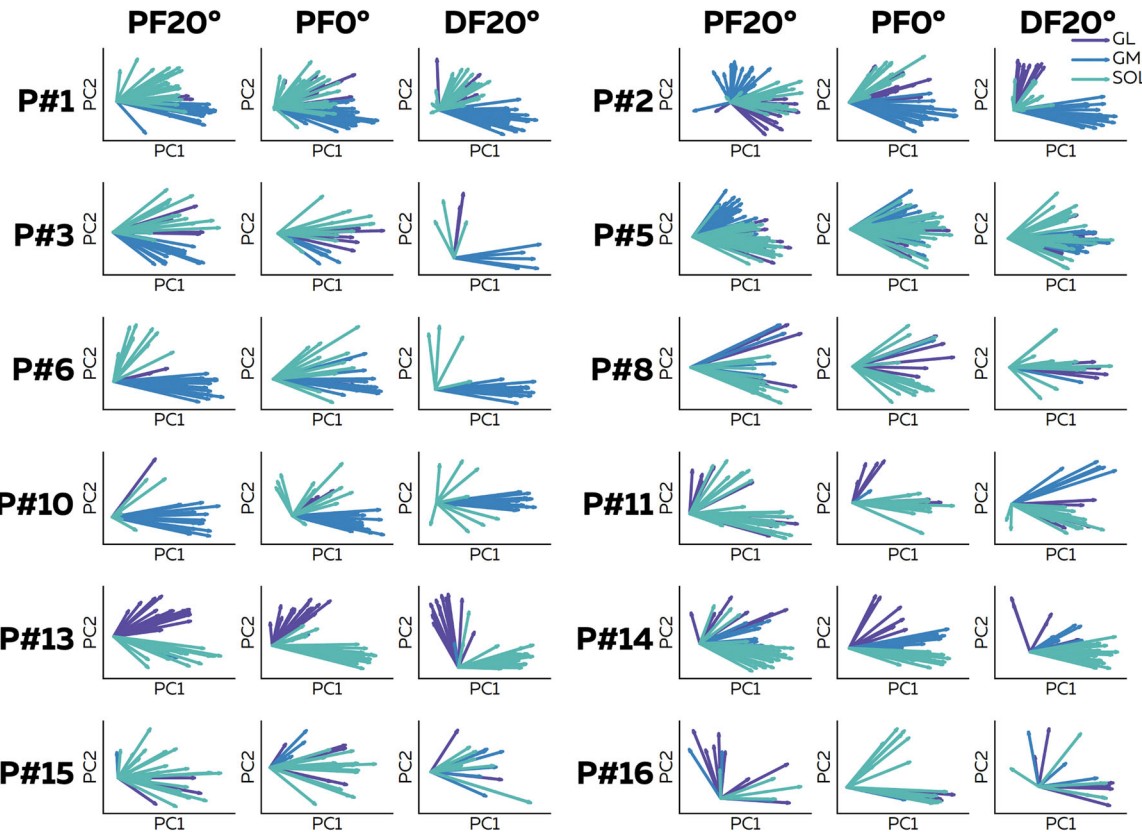

**Figure 4. Biplots of motor unit weights from the two motor neuron synergies**
For each participant ($n = 12$), we applied principal component analysis on normalized smoothed discharge rates concatenated in a matrix. The identified components were ranked based on the variance they explained, with two components sufficient such that the variance explained reached a plateau. Then, we displayed two-dimensional biplots, in which the coordinates of each motor neuron are the weights of the two motor neuron synergies. Each vector represents an individual motor neuron. The colour of the vector depends on the muscle. Abbreviations: DF, dorsiflexion; GL, gastrocnemius lateralis; GM, gastrocnemius medialis; PC, principal component; PF, plantarflexion; P#, participant number; and SOL, soleus. [Colour figure can be viewed at wileyonlinelibrary.com]

for GL–GM, GL–SOL and GM–SOL, respectively. The coefficients of correlation were significantly different between conditions ($F = 43.65$, $P < 0.001$) and muscles ($F = 141.46$, $P < 0.001$), with a significant interaction ($F = 15.14$, $P < 0.001$). Specifically, the coefficients of correlation for pairs of motor neurons within the GL and GM were significantly higher for PF0° ($P < 0.001$) and DF20° ($P < 0.001$) compared with PF20°. The coefficients for the SOL were not significantly different across ankle angles. The coefficients of correlation calculated between motor neurons of GL–GM, GL–SOL and GM–SOL were significantly higher for PF0° than for DF20° ($P = 0.012$, $P < 0.001$ and $P < 0.001$, respectively). The coefficients of correlation calculated between motor neurons of GL–SOL and GM–SOL were also significantly higher for PF0° than for PF20° ($P < 0.001$ and $P < 0.001$, respectively). The ratio of pairs of motor neurons with significant

correlations reached, on average, 98.0, 96.1 and 98.2% for GL, GM and SOL, respectively (Fig. 6*B*). The ratio of pairs of motor units with significant correlations reached, on average, 36.9, 78.2 and 90.2% for GL–GM, GL–SOL and GM–SOL, respectively.

We then compared the distribution of common inputs across ankle angles by calculating the ratio of pairs of motor neurons with the same level of correlation (either significant or non-significant correlations) across the three angles (Fig. 6*C*). The rest of the motor neurons switched between significant and non-significant correlations across ankle angles. None of the pairs of motor neurons had a non-significant level of correlation across all three ankle angles when considering motor neurons from GL, GM, SOL, GL–SOL and GM–SOL, but 26.1% of pairs of motor neurons from GL–GM had a non-significant level of correlation across all

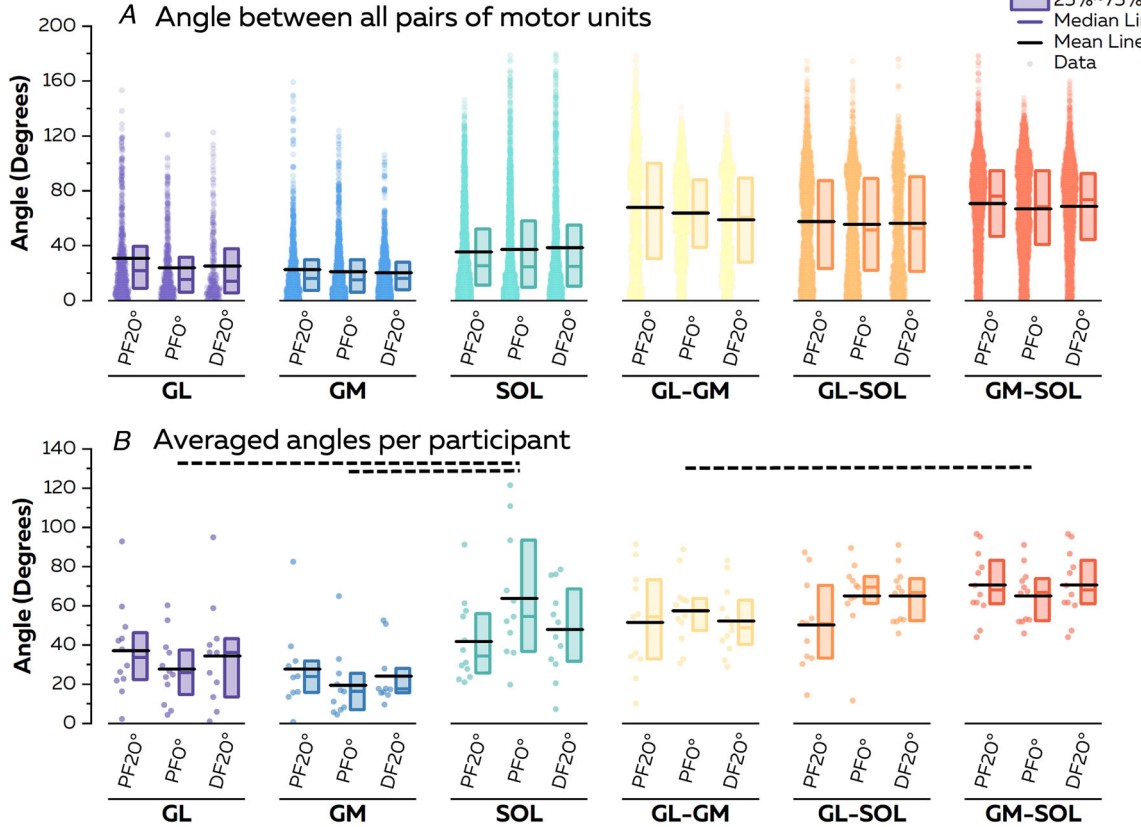

**Figure 5. Angles between vectors representing motor neurons in biplots**
After applying principal component analysis, we displayed two-dimensional biplots, in which coordinates of each motor neuron were the weights of that motor neuron in each of the two principal components. We then calculated the angles between these vectors to estimate the strength of common inputs received by each pair of motor neurons. *A*, we report the angle values for all the pairs of motor neurons. Each scatter represents an individual pair of motor neurons. *B*, we then averaged these values per participant and performed the statistical analyses to compare these values between muscles and conditions. Each scatter represents an individual. The horizontal dashed lines above the plot depict the significant statistical differences with $P < 0.05$. For the sake of clarity, note that we did not display the significant difference between GL and GM for all pairs of muscles. Abbreviations: DF, dorsiflexion; GL, gastrocnemius lateralis; GM, gastrocnemius medialis; PF, plantarflexion; and SOL, soleus. [Colour figure can be viewed at wileyonlinelibrary.com]

conditions. A high percentage of pairs of motor neurons had a significant level of correlation across all three ankle angles for GL (94.1%), GM (88.4%) and SOL (94.6%) (Fig. 6*C*). When considering pairs of motor neurons from different muscles, 13.0% (GL–GM), 48.3% (GL–SOL) and 73.5% (GM–SOL) had a consistent significant level of correlation across angles (Fig. 6*C*). It is noteworthy that we failed to identify a significant level of correlation in a large percentage of pairs of motor units between GL and GM. Conversely, some pairs of motor neurons from GL–SOL and GM–SOL exhibited significant correlations at all three angles. Together with the absence of significant correlations for most of the GM–GL pairs, this supports the findings from the PCA that two common inputs drive motor neurons innervating the GL, GM and SOL muscles during ankle plantarflexion.

## Discussion

The aim of this study was to test the robustness of motor neuron synergies across isometric plantarflexion tasks performed with different mechanical constraints. In the first part, we used simulated signals to estimate the sensitivity of our methods (i.e. PCA and correlation of smoothed discharge rates) to identify the number of common inputs and their distribution across motor neurons with various weightings. We found that PCA can accurately identify the number of common inputs to groups of motor neurons. Moreover, cross-correlations can accurately identify pairs of motor neurons receiving or not receiving a significant proportion of common inputs. This approach was then applied to experimental data. We identified two main common inputs driving motor neurons innervating the GL, GM and SOL, with no significant systematic changes in the distribution of

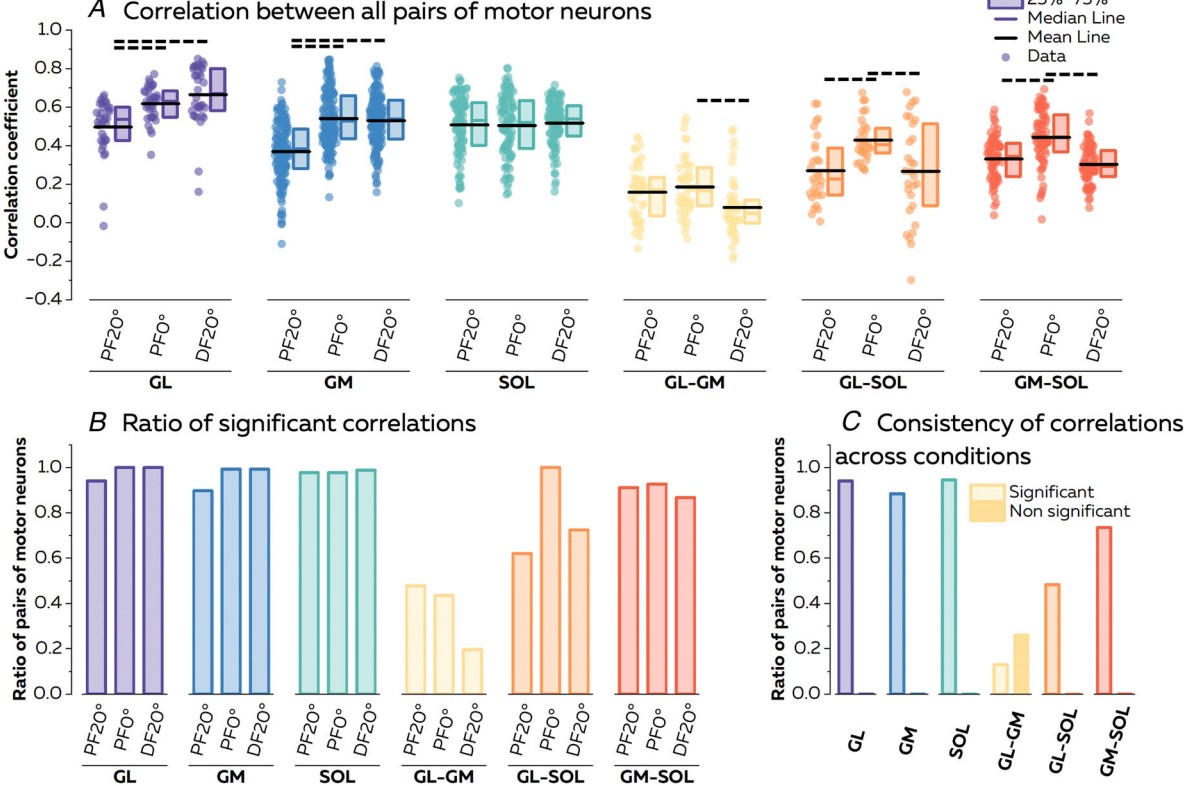

**Figure 6. Common inputs to pairs of motor neurons**
We estimated the proportion of common synaptic input for all pairs of motor neurons from the same and different muscles using cross-correlation analyses between the smoothed discharge rates of individual motor neurons. *A*, we first compared the level of correlation for each individual muscle and pairs of muscles. Each scatter represents a pair of motor neurons. The horizontal dashed lines above the plots depict the significant statistical differences with $P < 0.05$. For the sake of clarity, we report only the differences for the interaction. *B*, we calculated the ratio of pairs of motor neurons with significant correlations for each individual muscle and pairs of muscles. *C*, to compare the changes in common synaptic input to pairs of motor neurons across conditions, we calculated the ratio of pairs of motor neurons that exhibited consistent behaviour across ankle angles, i.e. either non-significant or significant level of correlation. Abbreviations: DF, dorsiflexion; GL, gastrocnemius lateralis; GM, gastrocnemius medialis; PF, plantarflexion; and SOL, soleus. [Colour figure can be viewed at wileyonlinelibrary.com]

weights across ankle angles. These results highlight a modular organization of the control of motor neurons innervating the triceps surae muscles with a reduced dimensionality.

## Methodological considerations

Our experiment is a first step towards demonstration of the modular organization of the neural control of motor neurons across tasks with different mechanical constraints. As previously noted, the assessment of motor neuron synergies across tasks requires the tracking of the same active motor neurons between contractions. To this end, we matched the unique spatiotemporal profiles of the action potentials recorded over the grid of electrodes (Del Vecchio & Farina, 2019; Martinez-Valdes et al., 2017; Oliveira & Negro, 2021). As changes in ankle position inevitably impacted the position and the angle of active muscle fibres relative to the electrodes, the profiles of action potentials varied slightly across conditions. Consequently, we had a significant reduction in the sample of motor units analysed across tasks (96 motor units, 17.2% of the total sample). The recent development of intramuscular electrode arrays (Chung et al., 2023; Muceli et al., 2022) might open new perspectives to accomplish this goal of matching motor units across motor tasks because these electrodes might be less affected by the changes in muscle geometry. Additionally, several teams have demonstrated the ability to decompose EMG signals during slow dynamic contractions, which might provide opportunities to show the robustness of motor neuron synergies during more natural behaviours (Chen et al., 2020; Glaser & Holobar, 2019; Oliveira & Negro, 2021).

Despite the above-mentioned methodological limitations and perspectives, studying the robustness of the distribution of common inputs driving the triceps surae muscles during plantarflexions performed at multiple angles has revealed relevant results. Previous works highlighted different changes in neural drive between the three muscles of the triceps surae during sustained contractions and in response to the alteration of force-generating capacity during an ankle rotation (Hug et al., 2021b; Lacourpaille et al., 2017; Rossato et al., 2022). Additionally, Marshall et al. (2022) have recently shown that the covariation of motor unit discharge rates and their recruitment order can change between short and long muscle lengths. Both results suggested that the CNS could drive these muscles with multiple common inputs, even during a task with a single degree of freedom. Our results supported this hypothesis, with the identification of two sources of common inputs driving the triceps surae muscles. Moreover, whether the distribution of common inputs to motor neurons is robust across muscle lengths was unknown. In this study, we also described the distribution of the two common inputs within and between muscles, showing a combination of the common inputs across motor neurons.

## Distribution of common inputs to modulate muscle force

Numerous studies have shown that most of the motor neurons from the same muscle display covariation of their discharge rate, which is necessary to modulate force during isometric contractions (Enoka & Farina, 2021; Farina & Negro, 2015; Farina et al., 2014b). These common fluctuations depend on the modulation of common synaptic inputs, received by all motor units (Farina & Negro, 2015; Farina et al., 2014a, b; Feeney et al., 2018; Negro et al., 2009). In line with these studies, we observed that none of the 273 pairs of motor units identified within GL, GM or SOL had a non-significant level of correlation for all three ankle angles (Fig. 6*C*). The covariation of discharge rates between motor units from either GL or GM was particularly high, with a ratio of pairs with significant correlation for all three ankle angles reaching 94.1 and 88.4% for GL and GM, respectively. This is in agreement with the small angles between vectors identified from the PCA (Fig. 5). Together, these observations indicate that the majority of motor neurons innervating either GL or GM received the two main common inputs with a similar weighting. On the contrary, the degree of covariation of discharge rates between motor neurons from GL and GM was lower than for the other pairs of muscles, with 13% of pairs of motor neurons exhibiting significant correlation across all three ankle angles. This is in agreement with previous work that used methods at the muscle level, such as intermuscular coherence (Hug et al., 2021b; Rossato et al., 2022) or PCA (Mazzo et al., 2022), to assess the covariations of motor unit discharge rates in the triceps surae muscles. Likewise, pairs of motor units from the SOL appear to have lower covariations of discharge rates than those from GL and GM. The nervous system might thus take advantage of the geometry and compartmentalization of this muscle (Bolsterlee et al., 2018) to control multiple sources of common input and produce forces in multiple directions (English et al., 1993; Segal et al., 1991). Such strategies have been observed in muscles controlling the individual fingers through multiple tendons (Hockensmith et al., 2005; Keen & Fuglevand, 2004; McIsaac & Fuglevand, 2007) or in the biceps brachii producing force for forearm supination and/or elbow flexion (Barry et al., 2009). Alternatively, one may consider the absence of correlation as a result of different non-linear input/output functions for each motor neuron owing to the activation of persistent inward currents (Binder et al., 2020; Heckman & Enoka, 2012; Heckman et al., 2008). However, we

focused our analysis on discharge rate oscillations at low frequencies, which are less affected by these non-linear transformations of synaptic inputs (Farina et al., 2014b; Negro & Farina, 2012). Therefore, we are confident that the absence of significant correlation identifies motor units receiving the two common inputs with very different weightings, as suggested for motor units from muscles of the upper limbs (Madarshahian et al., 2021; Marshall et al., 2022) and lower limbs (Del Vecchio et al., 2023; Hug et al., 2023b).

## From muscle to motor neuron synergies

The fact that the CNS might group motor units from synergistic muscles to reduce the dimensionality of movement control has been supported for decades by studies identifying muscle synergies (Bizzi & Cheung, 2013; Dominici et al., 2011; Takei et al., 2017; Ting et al., 2015; Tresch & Jarc, 2009). Several studies have shown similar levels of correlation between the discharge activities of motor neurons originating from two synergistic muscles as from the same muscle (Del Vecchio et al., 2023; Hug et al., 2023b). Thus, recent studies have proposed to decompose EMG signals to change the scale at which the synergies are described (Del Vecchio et al., 2023; Hug et al., 2023a, b; Madarshahian et al., 2021; Tanzarella et al., 2021). Here, PCA identified two principal components that explained the main variations of discharge rates of the identified motor neurons. It is worth noting that Gibbs et al. (1995) have already reported significant levels of synchronization between the discharge times of motor units from the gastrocnemius and the soleus muscles, consistent with our results outlining the projection of common inputs to the two muscles.

The results of this study support three of the hypotheses proposed by Hug et al. (2023a) within the motor neuron synergies framework, namely that during movement: (i) motor neurons receive common input in relatively large groups; (ii) motor neuron synergies might differ significantly from the classical definition of motor neuron pools, such that synergies might span across muscles (i.e. GL and SOL or GM and SOL) and/or involve only a portion of a muscle (i.e. SOL); and (iii) synergies represent motor modules used by the CNS to reduce the dimensionality of control. The last observation is the main novelty of our study. Previous studies identified motor neuron synergies during isometric contractions performed at a single joint angle (Del Vecchio et al., 2023; Hug et al., 2023b; Madarshahian et al., 2021) and, therefore, did not demonstrate their invariance across tasks, which is needed to prove a modular organization of movement control (d'Avella & Bizzi, 2005; d'Avella et al., 2003). To this end, we tested the robustness of motor neuron synergies across ankle angles, with no systematic changes in the distribution of principal component weights within and between muscles (Fig. 5) and with a large proportion of pairs of motor units consistently having significant and non-significant correlations between their smoothed discharge rates (Fig. 6C). Although a minority of pairs of motor units switched between significant and non-significant levels of correlations across ankle angles, our simulation showed that these motor units might receive the two sources of common inputs with relatively balanced weights (i.e. 34%/66% to 49%/51% in the simulation). The different changes in the amplitude of each source of common input might then impact the level of covariation between motor neuron discharge rates across angles. To support this idea, divergent variations in neural drive to the triceps surae muscles have been observed indirectly, while the position of the ankle changed during isometric contractions (Cibulka et al., 2017; Huang et al., 2016; Hug et al., 2021b; Lacourpaille et al., 2017). Overall, we suggest that the distribution of common inputs across motor neurons is indeed robust across tasks, with some changes in level of correlation attributable to differential changes in the amplitude of inputs that the motor neurons receive.

The concept of combining motor neuron synergies to generate movements might bridge the gap between studies supporting modular control and flexible control of motor units. Proponents of flexible control of motor units suggest that motor units from the same muscle might receive independent inputs instead of one common input, explaining changes in recruitment order between the same motor units across multiple tasks (Basmajian, 1963; Formento et al., 2021; Marshall et al., 2022). We show here that most of the motor units from the triceps surae muscles might receive two sources of common inputs. The combination of multiple independent inputs might therefore explain why previous works reported changes in recruitment strategies between motor units from the same muscle while changing the mechanical constraints of the task (Desnedt & Gidaux, 1981; Herrmann & Flanders, 1998; Marshall et al., 2022; ter Haar Romeny et al., 1984).

## Neural structures that shape the modular control of calf muscles

These results suggest that the neural control of movements might be organized hierarchically, whereby motor modules are controlled instead of individual motor neurons (Loeb et al., 1999). Thus, spinal circuits involving premotor neurons and interneurons might span across motor nuclei to transmit these correlated inputs to synergist muscles (Levine et al., 2014; Ronzano et al., 2021; Takei et al., 2017). However, given that motor neurons are the final common pathways where inputs

from supraspinal centres and afferent fibres converge, we cannot make any inference about the origin of the correlated inputs, although several candidates exist. For example, previous studies found that stimulation of the motor cortex (Overduin et al., 2012) or premotor neurons in the spinal cord (Takei et al., 2017) evoked synergies comparable to those observed during natural behaviours. Additionally, Cheung et al. (2005) observed similar synergies in animals with intact and deafferented nervous systems, suggesting that the activation of synergies would mostly be a feedforward process. Alternatively, others proposed that the correlated activity between synergistic muscles might aim to regulate mechanical features, such as internal joint stress, thus emphasizing the key role of sensory feedback (Alessandro et al., 2020). This could also be the case in our study, because the variation of ankle angles undoubtedly impacts the behaviour of motor neurons during plantarflexions (Kennedy & Cresswell, 2001; Trypidakis et al., 2022). Additionally, the two motor modules might allow the CNS to tune the behaviour of the lateral and medial portions of the calf muscles to stabilize the foot, producing plantarflexion in the frontal plane (Lee & Piazza, 2008), thus limiting the strain and the stress of the ankle joint. Overall, future work is needed to identify the neural bases of common inputs to motor neurons during voluntary movements (Cheung & Seki, 2021).

## Conclusion

Our results support the idea that the CNS might combine motor modules to produce ankle plantarflexion. Specifically, we identified two motor neuron synergies grouping motor neurons innervating the triceps surae muscles during plantarflexion performed at multiple ankle angles. Importantly, we found that the structure of these motor neuron synergies is mostly invariant across conditions despite the variation of mechanical constraints, with some level of flexibility that might be accounted for by differential changes in the amplitude of each input. This provides a new observation that contributes to demonstrate the functional relevance of such modular structures to generate movements.

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

## Additional information

### Data availability statement

Data and codes will be made available upon request to the authors.

### Competing interests

The authors declare no competing financial interests.

### Author contributions

J.L.: Conception or design of the work; Acquisition, analysis or interpretation of data for the work; Drafting the work or revising it critically for important intellectual content; Final approval of the version to be published; Agreement to be accountable for all aspects of the work. S.A.: Conception or design of the work; Acquisition, analysis or interpretation of data for the work; Drafting the work or revising it critically for important intellectual content; Final approval of the version to be published; Agreement to be accountable for all aspects of the work. D.F.: Acquisition, analysis or interpretation of data for the work; Drafting the work or revising it critically for important intellectual content; Final approval of the version to be published; Agreement to be accountable for all aspects of the work. F.H.: Acquisition, analysis or interpretation of data for the work; Drafting the work or revising it critically for important intellectual content; Final approval of the version to be published; Agreement to be accountable for all aspects of the work. J.L.P.: Acquisition, analysis or interpretation of data for the work; Drafting the work or revising it critically for important intellectual content; Final approval of the version to be published; Agreement to be accountable for all aspects of the work.

### Funding

Dario Farina is supported by the European Research Council Synergy Grant NaturalBionicS (contract no. 810346), the EPSRC Transformative Healthcare, NISNEM Technology (EP/T020970) and the BBSRC, 'Neural Commands for Fast Movements in the Primate Motor System' (NU-003743). François Hug is supported by a fellowship from the French government, through the UCAJEDI Investments in the Future and by the National Research Agency (ANR), with the reference number ANR-15-IDEX-01. Jose L. Pons is supported by the National Science Foundation/National Robotics Initiative under grant no. 2024488.

### Acknowledgements

We thank Allie Levine for her illustration in Fig. 1.

### Keywords

motor control, motor modules, motor neuron, synergy

## Supporting information

Additional supporting information can be found online in the Supporting Information section at the end of the HTML view of the article. Supporting information files available:

**Statistical Summary Document**
**Peer Review History**

