## [Peer Review History · The Journal of Physiology]

Two motor neuron synergies, invariant across ankle joint angles, activate the triceps surae during plantarflexion

Jackson Levine, Simon Avrillon, Dario Farina, Francois Hug, and José L. Pons
DOI: 10.1113/JP284503

Corresponding author(s): Simon Avrillon (s.avrillon@imperial.ac.uk)

The following individual(s) involved in review of this submission have agreed to reveal their identity: Andrea d'Avella (Referee #1); Vincent Cheung (Referee #2)

Review Timeline:

Submission Date:	05-Feb-2023
Editorial Decision:	24-Apr-2023
Revision Received:	26-Jun-2023
Editorial Decision:	20-Jul-2023
Revision Received:	03-Aug-2023
Accepted:	10-Aug-2023

Senior Editor: Richard Carson

Reviewing Editor: Madeleine Lowery

Transaction Report:

Dear Dr Avrillon,

Re: JP-RP-2023-284503 "Two motor neuron synergies, invariant across ankle joint angles, activate the triceps surae during plantarflexion" by Jackson Levine, Simon Avrillon, Dario Farina, Francois Hug, and José L. Pons

Thank you for submitting your manuscript to The Journal of Physiology. It has been assessed by a Reviewing Editor and by 2 expert referees and we are pleased to tell you that it is potentially acceptable for publication following satisfactory major revision.

REVISION CHECKLIST:

We look forward to receiving your revised submission.

Yours sincerely,

Richard Carson
Senior Editor
The Journal of Physiology

REQUIRED ITEMS

- Author photo and profile. First (or joint first) authors are asked to provide a short biography (no more than 100 words for one author or 150 words in total for joint first authors) and a portrait photograph. These should be uploaded and clearly labelled with the revised version of the manuscript. See Information for Authors for further details.
- You must start the Methods section with a paragraph headed Ethical Approval. If experiments were conducted on humans confirmation that informed consent was obtained, preferably in writing, that the studies conformed to the standards set by the latest revision of the Declaration of Helsinki, and that the procedures were approved by a properly constituted ethics committee, which should be named, must be included in the article file. If the research study was registered (clause 35 of the Declaration of Helsinki) the registration database should be indicated, otherwise the lack of registration should be noted as an exception (e.g. The study conformed to the standards set by the Declaration of Helsinki, except for registration in a database). For further information see: <https://physoc.onlinelibrary.wiley.com/hub/human-experiments>
- The Journal of Physiology funds authors of provisionally accepted papers to use the premium BioRender site to create high resolution schematic figures. Follow this link and enter your details and the manuscript number to create and download figures. Upload these as the figure files for your revised submission. If you choose not to take up this offer we require figures to be of similar quality and resolution. If you are opting out of this service to authors, state this in the Comments section on the Detailed Information page of the submission form. The link provided should only be used for the purposes of this submission. Authors will be charged for figures created on this premium BioRender account if they are not related to this manuscript submission.
- Please upload separate high-quality figure files via the submission form.
- Please ensure that any tables are in Word format and are, wherever possible, embedded in the article file itself.
- Please ensure that the Article File you upload is a Word file.
- A Statistical Summary Document, summarising the statistics presented in the manuscript, is required upon revision. It must be on the Journal's template, which can be downloaded from the link in the Statistical Summary Document section here: https://jp.msubmit.net/cgi-bin/main.plex?form_type=display_requirements#statistics.
- Papers must comply with the Statistics Policy: https://jp.msubmit.net/cgi-bin/main.plex?form_type=display_requirements#statistics.

In summary:

- If $n \leq 30$, all data points must be plotted in the figure in a way that reveals their range and distribution. A bar graph with data points overlaid, a box and whisker plot or a violin plot (preferably with data points included) are acceptable formats.
- If $n > 30$, then the entire raw dataset must be made available either as supporting information, or hosted on a not-for-profit repository e.g. FigShare, with access details provided in the manuscript.
- 'n' clearly defined (e.g. x cells from y slices in z animals) in the Methods. Authors should be mindful of pseudoreplication.
- All relevant 'n' values must be clearly stated in the main text, figures and tables, and the Statistical Summary Document (required upon revision).
- The most appropriate summary statistic (e.g. mean or median and standard deviation) must be used. Standard Error of the

Mean (SEM) alone is not permitted.

- Exact p values must be stated. Authors must not use 'greater than' or 'less than'. Exact p values must be stated to three significant figures even when 'no statistical significance' is claimed.

- Statistics Summary Document completed appropriately upon revision.

- Please include an Abstract Figure file, as well as the figure legend text within the main article file. The Abstract Figure is a piece of artwork designed to give readers an immediate understanding of the research and should summarise the main conclusions. If possible, the image should be easily 'readable' from left to right or top to bottom. It should show the physiological relevance of the manuscript so readers can assess the importance and content of its findings. Abstract Figures should not merely recapitulate other figures in the manuscript. Please try to keep the diagram as simple as possible and without superfluous information that may distract from the main conclusion(s). Abstract Figures must be provided by authors no later than the revised manuscript stage and should be uploaded as a separate file during online submission labelled as File Type 'Abstract Figure'. Please ensure that you include the figure legend in the main article file. All Abstract Figures should be created using BioRender. Authors should use The Journal's premium BioRender account to export high-resolution images. Details on how to use and access the premium account are included as part of this email.

EDITOR COMMENTS

Reviewing Editor:

Both reviewers recognize the contribution of the manuscript and have raised a number of issues which should be addressed before it is ready for publication.

Senior Editor:

As you will note, the Referees were positively disposed towards the intent and the execution of the project. As also emphasised by the Reviewing Editor however, there are certain aspects of the presentation that detract from the potential impact of this work. Most conspicuous among these is the application of a clustering algorithm to the motor unit weights. It is emphasised by both referees that neither the logic nor the conceptual basis for this step appear convincing. In the event that you opt to revise your submission, I would ask that you pay particular attention to this issue.

REFEREE COMMENTS

Referee #1:

This study investigated the robustness of motor neuron synergies, i.e., groups of motor neurons belonging to motor neuron pools of multiple muscles and sharing the same common input, across task conditions. First, the use of principal component analysis (PCA) on the smoothed discharge rates of multiple motor units to identify the number of independent common inputs and of cross-correlation to identify pairs of motor units receiving the same common input were validated using simulated data. Then, PCA and cross-correlation analyses were applied to the smoothed discharge rates of motor units identified in gastrocnemius lateralis (GL), gastrocnemius medialis (GM), and soleus (SOL) from high-density surface electromyographic recordings during generation of isometric plantarflexion torque at three different ankle angles. Even if the task involved a single degree of freedom, PCA allowed to identify two motor neuron synergies with most of the motor units assigned to each synergy maintaining a significant cross-correlation across angles.

The investigation of motor neuron synergies is critical to gain insights in the neural implementation of a modular control architecture. This study is novel and important because it demonstrates for the first time the robustness of motor neuron synergies across different task conditions, supporting the hypothesis that such synergies represent a set of invariant modules that may reduce the computational burden of movement control. There is, however, one key aspect of the conceptual framework used to interpret the results and to define one of the steps of the analysis of both simulated and experimental data that is unnecessary and potentially misleading.

Classifying each motor neuron as belonging to one motor neuron synergies, and consequently applying a clustering algorithm on the weights of each motor unit in the two first principal components, is justified under the assumption that each motor neuron belongs to only one synergy. While this is a reasonable possibility, there is no reason not to also consider the possibility that each motor neurons may belong to more than one synergy. Indeed, this possibility is implemented in the simulation, where 60% of the motor neurons received both common inputs. Then, if each motor neurons may have synaptic weights of different magnitudes for both common inputs, there is no need and no justification for a classification of each motor neuron. If the representation of the motor neurons in the two-dimensional space of synaptic weights (or equivalently of the weight of the two principal components) is not concentrated in specific portions of the space, clustering becomes arbitrary. Indeed, in many of the cases illustrated in Fig. 4 (e.g., for P#1, P#5, P#14), the distribution of the motor units appears rather uniform in a broad region of space. Clearly one can apply a clustering algorithm to a uniform distribution and

classify its elements, but this operation likely does not provide any additional information on the property of that distribution. In fact, the key insight and important result of the study, the identification of two motor unit synergies, does not depend on the potentially unsupported observation of distinct clusters.

On the contrary, considering the fact that motor neurons may belong to more than one motor neuron synergies, makes motor neuron synergies a plausible model for the implementation of muscle synergies and avoids presenting motor neuron synergies and muscle synergies as two contrasting views. Synergies defined at the muscle levels do not prescribe any specific neural implementation, as muscle synergy models describe the generation of muscle patterns at a functional rather than at a neural implementation level. Thus, synergies at the muscle level do not require the "implicit assumption" that "all motor neurons innervating a muscle (i.e., a motor neuron pool) receive the same common inputs" (lines 88-89, 631-632), as they do not model the generation of muscle patterns at the motor unit level.

Referee #2:

In this study, the authors seek to uncover invariant patterns of motoneuronal co-activations across the three muscles belonging to the triceps surae (medial (GM) and lateral (GL) heads of gastrocnemius and soleus (SOL)). Prior studies of motor modularity usually involve analysis of multi-muscle EMGs with muscle synergies defined at the whole-muscle level. Here, by uncovering common inputs to motor units belonging to potentially different muscles, the authors extend the motor modularity concept to the motoneuronal level. High-density EMGs (64 channels) were recorded during an isometric task with the ankle fixed at three different angles. After the motor unit spikes were decomposed from the hd-EMG, a procedure based on the principal component analysis (PCA), validated by simulations, was used to reveal how multiple motor units were co-activated by common inputs. The authors found two motoneuronal synergies, involving GL-SOL and GM-SOL, that recruited the same motoneurons across the three ankle angles. These synergies are interpreted to be plausible modules of lower-limb motor control.

Overall, this is a very well written paper that addresses a timely, important question in motor neuroscience. It demonstrates how motor modularity may be productively studied at the motoneuronal level. The methods adopted here may also become a framework for later hd-EMG studies that investigate motoneuronal synergies. But to further enhance the paper's potential impact, the authors are suggested to further clarify the following points in their revision.

MAJOR COMMENTS

(1) To identify the motoneuronal synergies from the smoothed discharge rates of the motor unit spikes, the authors relied on PCA, which necessarily assumes that each motor unit can be under the influence of more than one common input, and that this influence may either be excitatory or inhibitory. But then, after PCA, the authors performed an additional step of plotting the weights of every motor unit of the first two principal components on x and y coordinates, and then applied k-means on this plane to cluster each motor unit into a single motoneuronal synergy (lines 313-315). This step essentially forces each motor unit to receive a single common input, thus making the PCA strength of allowing each data channel to be contributed by multiple sources appearing superfluous. These analytic steps seem all the more curious when the authors explicitly noted that the assignment of motor units to clusters became inconsistent when the units receive balanced activations from multiple common inputs (e.g., lines 382-383).

Importantly, in light of the above methodological choices, the statement "74% of the motor units belonged to the same clusters across all the conditions" (lines 477-479) should be qualified with the caveat that synergy robustness was observed only after the additional step of assigning each motor unit into a single synergy.

I suggest reporting the actual principal components (PCs) (i.e., the eigenvectors) and presenting them in a way that shows the correspondence between the motor-unit weights across the components (e.g., show the PCs as bar graphs). Robustness of the motoneuronal synergies can then potentially be demonstrated through high similarity values between the PC vectors from the different conditions. If the k-means step is retained, the authors should better justify why this is done. If each motor unit can belong only to a single synergy, then why not just apply clustering on the spike discharge rates or rely solely on correlation measures?

(2) After performing PCA, the authors presented another analysis based on pairwise correlation of motor-unit discharge rates to further strengthen the observations of GL-SOL and GM-SOL synergies as demonstrated in the PCA analysis. Since each motor unit can be activated by multiple and different common inputs (as assumed in the authors' simulations), the lack of correlations between any two motor units does not necessarily imply that they are not co-activated in any one of the motoneuronal synergy. Correlation-based analysis is thus "less sophisticated" in a way than PCA-based analysis, because the former implicitly assumes that each motor unit belongs only to one synergy.

I can see the value of including correlation results in this work, but the authors can better justify why this step is included after PCA. I suggest using the correlation results to benchmark the PCA results, especially in the simulation part. To what extent can co-activation patterns be revealed only by PCA but not by correlations?

(3) The authors pointed out one potential limitation of the classic muscle synergy model (lines 88-89; 630-632) in that it assumes that all motoneurons innervating a muscle receive the same common inputs. I can see this as one possible

interpretation of the classic model (which does not appear to be explicitly acknowledged in Cheung & Seki 2021, as claimed on line 632). But it is also true that the classic model aims only to describe the EMGs of individual muscles at the whole-muscle level, and has nothing to say concerning which motor units are recruited in what order whenever the muscle is activated through one or multiple synergies. Very importantly, the classic model does not need this "implicit assumption" to work. For instance, in the classic model, the EMG of a muscle can be the summation of contributions from two different muscle synergies, and each synergy may recruit two different collections of motor units belonging to this same muscle to result in the final EMG of this muscle.

MINOR COMMENTS

- (1) Lines 112-113: Can clarify that hd-EMGs were recorded during isometric conditions at three different ankle angles.
- (2) Lines 268-270: Can further clarify how each isolated motor unit was attributed to an individual muscle (GM, GL, or SOL). Was this based solely on the anatomical positions of the channels that contributed to the isolated unit?
- (3) Lines 277-278: Can clarify what exactly was averaged through spike-triggered averaging.
- (4) Lines 320-321: It is a bit unclear why the criterion stated here is good for finding the 10-s window for cross-correlation.
- (5) Fig. 2 & Fig. 4: Were the signs of any identified PCs flipped according to a certain criterion?
- (6) Line 439: It would be informative to report the average and range of the number of motor units identified per subject, for each of the three ankle angles.

END OF COMMENTS

Confidential Review

05-Feb-2023

EDITOR COMMENTS

Reviewing Editor:

Both reviewers recognize the contribution of the manuscript and have raised a number of issues which should be addressed before it is ready for publication.

Senior Editor:

As you will note, the Referees were positively disposed towards the intent and the execution of the project. As also emphasised by the Reviewing Editor however, there are certain aspects of the presentation that detract from the potential impact of this work. Most conspicuous among these is the application of a clustering algorithm to the motor unit weights. It is emphasised by both referees that neither the logic nor the conceptual basis for this step appear convincing. In the event that you opt to revise your submission, I would ask that you pay particular attention to this issue.

Dear editors and reviewers, we thank you for your feedback and your comments that helped us to improve the manuscript. Please note that in addition to addressing your comments, we improved some analyses and updated the associated results. Specifically, motor units are now matched using single-differential EMG signals instead of EMG signals recorded with a monopolar montage. We believe that this new analysis increases the accuracy of the matching by sampling the spatial distribution of motor unit action potentials within the grid with greater spatial selectivity. Additionally, we ran the principal component analysis on all the identified motor units instead of matched motor units only. By doing so, we increased the statistical power of the analysis by increasing the number of participants (12 instead of 7) and the total number of identified motor units (34 ± 17 instead of 15 ± 7 per participant) considered in the analysis. We believe that our conclusions are strengthened by the updated results.

All comments and suggestions of the reviewers have been taken into account in this revision, as detailed in the point-by-point replies provided in the following.

REFeree COMMENTS

Referee #1:

This study investigated the robustness of motor neuron synergies, i.e., groups of motor neurons belonging to motor neuron pools of multiple muscles and sharing the same common input, across task conditions. First, the use of principal component analysis (PCA) on the smoothed discharge rates of multiple motor units to identify the number of independent common inputs and of cross-correlation to identify pairs of motor units receiving the same common input were validated using simulated data. Then, PCA and cross-correlation analyses were applied to the smoothed discharge rates of motor units identified in gastrocnemius lateralis (GL), gastrocnemius medialis (GM), and soleus (SOL) from high-density surface electromyographic recordings during generation of isometric plantarflexion torque at three different ankle angles. Even if the task involved a single degree of freedom, PCA allowed to identify two motor neuron synergies with most of the motor units assigned to each synergy maintaining a significant cross-correlation across angles.

The investigation of motor neuron synergies is critical to gain insights in the neural implementation of a modular control architecture. This study is novel and important because it demonstrates for the first time the robustness of motor neuron synergies across different task conditions, supporting the hypothesis that such synergies represent a set of invariant modules that may reduce the computational burden of movement control. There is, however, one key aspect of the conceptual framework used to interpret the results and to define one of the steps of the analysis of both simulated and experimental data that is unnecessary and potentially misleading.

Classifying each motor neuron as belonging to one motor neuron synergies, and consequently applying a clustering algorithm on the weights of each motor unit in the two first principal components, is justified under the assumption that each motor neuron belongs to only one synergy. While this is a reasonable possibility, there is no reason not to also consider the possibility that each motor neurons may belong to more than one synergy. Indeed, this possibility is implemented in the simulation, where 60% of the motor neurons received both common inputs. Then, if each motor neurons may have synaptic weights of different magnitudes for both common inputs, there is no need and no justification for a classification of each motor neuron. If the representation of the motor neurons in the two-dimensional space of synaptic weights (or equivalently of the weight of the two principal components) is not concentrated in specific portions of the space, clustering becomes arbitrary. Indeed, in many of the cases illustrated in Fig. 4 (e.g., for P#1, P#5, P#14), the distribution of the motor units appears rather uniform in a broad region of space. Clearly one can apply a clustering algorithm to a uniform distribution and classify its elements, but this operation likely does not provide any additional information on the property of that distribution. In fact, the key insight and important result of the study, the identification of two motor unit synergies, does not depend on the potentially unsupported observation of distinct clusters.

We thank the reviewer for this important insight. We agree that motor neurons could receive multiple inputs from common drivers. This is indeed at the basis of the concept of synergies at the motor neuron level. The reviewer is right that clustering the weights of the motor neurons can be misleading. To address this comment, we have removed the clustering analysis and we now focus our results on the principal component analysis applied on all the identified motor units. Despite considering more motor neurons and participants than in the original version (see general comment above), we identified the same number of synergies, i.e., two. To replace the analysis based on the clustering, we have calculated the angle between vectors displayed in the two-dimensional biplots. As shown with the simulation, the larger the difference between the distribution of correlated inputs to two motor neurons, the larger the angle between the two vectors. The adjusted R^2 of linear fits between these angles and the differences in ratio of common input 1 ranged between 0.91 and 0.92 for the three conditions of our simulation.

We observed a significant main effect of muscle on experimental data. Specifically, our analysis highlights that the angles between motor neurons innervating the GL or the GM are significantly smaller than those between motor neurons innervating the SOL. Additionally, we did not observe a significant main effect of the ankle angle nor an interaction between angle and muscles. This confirms our hypothesis that the two synergies mostly separate

motor units from GL and GM across ankle angles, while motor neurons from the SOL are spread within the biplots.

The manuscript and the figures 2, 4 and 5 have been updated.

On the contrary, considering the fact that motor neurons may belong to more than one motor neuron synergies, makes motor neuron synergies a plausible model for the implementation of muscle synergies and avoids presenting motor neuron synergies and muscle synergies as two contrasting views. Synergies defined at the muscle levels do not prescribe any specific neural implementation, as muscle synergy models describe the generation of muscle patterns at a functional rather than at a neural implementation level. Thus, synergies at the muscle level do not require the "implicit assumption" that "all motor neurons innervating a muscle (i.e., a motor neuron pool) receive the same common inputs" (lines 88-89, 631-632), as they do not model the generation of muscle patterns at the motor unit level.

The reviewer is right. We updated both the introduction and the discussion sections to realign the frameworks of muscle and motor neuron synergies. Thus, motor neuron synergies are presented as a change of scale to describe the distribution of common inputs across groups of motor neurons that enables the nervous system to reduce the dimension of the neural control of movement.

You can now read:

(P.3, L.84): "This concept of synergies (or motor modules) has been mainly demonstrated at the muscle level by combining the recording of multiple muscles with bipolar electromyography and factorization algorithms (Tresch et al., 1999; d'Avella et al., 2003; Ting et al., 2015; Yaron et al., 2020). The recent development of algorithms that identify the discharge activity of active motor units revealed a new dimensionality of movement control, where synergies are considered at the motor neuron level. These motor neuron synergies can cover subsets of motor neurons within each muscle or across portions of different muscles (Hug et al., 2022)."

(P.25, L.622): "The fact that the central nervous system may group motor units from synergistic muscles to reduce the dimensionality of the neural control of movements has been supported for decades by the theory of muscle synergies (Tresch & Jarc, 2009; Dominici et al., 2011; Bizzi & Cheung, 2013; Ting et al., 2015; Takei et al., 2017). Of note, these synergies have been described at the muscle level. However, several studies have highlighted similar levels of correlation between the firing activities of motor units originating from two synergistic muscles as from the same muscle (Del Vecchio et al., 2022; Hug et al., 2022b). Thus, recent studies have proposed to take advantage of the large sampling of motor unit activity from decomposed HD-EMG signals to extend the synergy concept to the motor neuron level (Madarshahian et al., 2021; Tanzarella et al., 2021; Del Vecchio et al., 2022; Hug et al., 2022; Hug et al., 2023)."

Referee #2:

In this study, the authors seek to uncover invariant patterns of motoneuronal co-activations across the three muscles belonging to the triceps surae (medial (GM) and lateral (GL) heads of gastrocnemius and soleus (SOL)). Prior studies of motor modularity usually involve analysis of multi-muscle EMGs with muscle synergies defined at the whole-muscle level. Here, by uncovering common inputs to motor units belonging to potentially different muscles, the authors extend the motor modularity concept to the motoneuronal level. High-density EMGs (64 channels) were recorded during an isometric task with the ankle fixed at three different angles. After the motor unit spikes were decomposed from the hd-EMG, a procedure based on the principal component analysis (PCA), validated by simulations, was used to reveal how multiple motor units were co-activated by common inputs. The authors found two motoneuronal synergies, involving GL-SOL and GM-SOL, that recruited the same motoneurons across the three ankle angles. These synergies are interpreted to be plausible modules of lower-limb motor control.

Overall, this is a very well written paper that addresses a timely, important question in motor neuroscience. It demonstrates how motor modularity may be productively studied at the motoneuronal level. The methods adopted here may also become a framework for later hd-EMG studies that investigate motoneuronal synergies. But to further enhance the paper's potential impact, the authors are suggested to further clarify the following points in their revision.

MAJOR COMMENTS

(1) To identify the motoneuronal synergies from the smoothed discharge rates of the motor unit spikes, the authors relied on PCA, which necessarily assumes that each motor unit can be under the influence of more than one common input, and that this influence may either be excitatory or inhibitory. But then, after PCA, the authors performed an additional step of plotting the weights of every motor unit of the first two principal components on x and y coordinates, and then applied k-means on this plane to cluster each motor unit into a single motoneuronal synergy (lines 313-315). This step essentially forces each motor unit to receive a single common input, thus making the PCA strength of allowing each data channel to be contributed by multiple sources appearing superfluous. These analytic steps seem all the more curious when the authors explicitly noted that the assignment of motor units to clusters became inconsistent when the units receive balanced activations from multiple common inputs (e.g., lines 382-383).

Importantly, in light of the above methodological choices, the statement "74% of the motor units belonged to the same clusters across all the conditions" (lines 477-479) should be qualified with the caveat that synergy robustness was observed only after the additional step of assigning each motor unit into a single synergy.

I suggest reporting the actual principal components (PCs) (i.e., the eigenvectors) and presenting them in a way that shows the correspondence between the motor-unit weights across the components (e.g., show the PCs as bar graphs). Robustness of the motoneuronal synergies can then potentially be demonstrated through high similarity values between the PC vectors from the different conditions. If the k-means step is retained, the authors should better justify why this is done. If each motor unit can belong only to a single synergy, then

why not just apply clustering on the spike discharge rates or rely solely on correlation measures?

We thank the reviewer for this important insight. As reported in the first response to reviewer 1, we removed the clustering analysis to make the results more consistent with the framework of muscle/motor neuron synergies. We applied PCA on the full sample of identified motor units in the new version of the manuscript (see general comment above). Therefore, it is not possible to directly compare the distribution of PC vectors across ankle angles, as the identified motor units may change. To replace the analysis based on the clustering, we have calculated the angle between vectors displayed in the two-dimensional biplots. As shown with the simulation, the larger the difference between the distribution of correlated inputs to two motor neurons, the larger the angle between the two vectors representing these two motor neurons on the biplot. The adjusted R^2 of linear fits between these angles and the differences in ratio of common input 1 ranged between 0.91 and 0.92 for the three conditions of our simulation.

We observed a significant main effect of the muscle on experimental data. Specifically, our analysis highlights that the angles between motor neurons innervating the GL or the GM are significantly smaller than those between motor neurons innervating the SOL. Additionally, we did not observe a significant main effect of the angle nor an interaction between angle and muscles.

The updated results are reported in the method and result sections, and in figures 2, 4 and 5.

(2) After performing PCA, the authors presented another analysis based on pairwise correlation of motor-unit discharge rates to further strengthen the observations of GL-SOL and GM-SOL synergies as demonstrated in the PCA analysis. Since each motor unit can be activated by multiple and different common inputs (as assumed in the authors' simulations), the lack of correlations between any two motor units does not necessarily imply that they are not co-activated in any one of the motoneuronal synergy. Correlation-based analysis is thus "less sophisticated" in a way than PCA-based analysis, because the former implicitly assumes that each motor unit belongs only to one synergy.

I can see the value of including correlation results in this work, but the authors can better justify why this step is included after PCA. I suggest using the correlation results to benchmark the PCA results, especially in the simulation part. To what extent can co-activation patterns be revealed only by PCA but not by correlations?

In the new version of the manuscript, the cross-correlation analysis is reported for matched motor units, which adds value to this analysis. It allows us to determine whether the distribution of common inputs is stable between tasks. Specifically, we considered that two motor units with a consistent significant correlation between their smoothed discharge rates across the three ankle angles are likely to receive a significant level of common inputs. On the contrary, if two motor units exhibit consistent non-significant correlation, regardless of the ankle angle, or exhibit variable levels of correlation depending on the ankle angle, they are likely to mostly receive independent inputs, as supported by our simulation. We believe that this new analysis strengthens the observation made with the PCA.

These updated results are reported in the result sections, and in figure 6.

(3) The authors pointed out one potential limitation of the classic muscle synergy model (lines 88-89; 630-632) in that it assumes that all motoneurons innervating a muscle receive the same common inputs. I can see this as one possible interpretation of the classic model (which does not appear to be explicitly acknowledged in Cheung & Seki 2021, as claimed on line 632). But it is also true that the classic model aims only to describe the EMGs of individual muscles at the whole-muscle level, and has nothing to say concerning which motor units are recruited in what order whenever the muscle is activated through one or multiple synergies. Very importantly, the classic model does not need this "implicit assumption" to work. For instance, in the classic model, the EMG of a muscle can be the summation of contributions from two different muscle synergies, and each synergy may recruit two different collections of motor units belonging to this same muscle to result in the final EMG of this muscle.

Our interpretation of Fig 2E from Cheung and Seki (2021) was that inputs are distributed to the whole motor neuron pool. But we agree with the reviewer, and as reported in the second response to reviewer 1, we rephrased the introduction and discussion sections to better align the framework of muscle and motor neuron synergies. We now present the motor neuron synergies as a change of scale to describe the low dimensional control of groups of motor units.

You can now read:

(P.3, L.84): *“This concept of synergies (or motor modules) has been mainly demonstrated at the muscle level by combining the recording of multiple muscles with bipolar electromyography and factorization algorithms (Tresch et al., 1999; d’Avella et al., 2003; Ting et al., 2015; Yaron et al., 2020). The recent development of algorithms that identify the discharge activity of active motor units revealed a new dimensionality of movement control, where synergies are considered at the motor neuron level. These motor neuron synergies can cover subsets of motor neurons within each muscle or across portions of different muscles (Hug et al., 2022).”*

(P.25, L.622): *“The fact that the central nervous system may group motor units from synergistic muscles to reduce the dimensionality of the neural control of movements has been supported for decades by the theory of muscle synergies (Tresch & Jarc, 2009; Dominici et al., 2011; Bizzi & Cheung, 2013; Ting et al., 2015; Takei et al., 2017). Of note, these synergies have been described at the muscle level. However, several studies have highlighted similar levels of correlation between the firing activities of motor units originating from two synergistic muscles as from the same muscle (Del Vecchio et al., 2022; Hug et al., 2022b). Thus, recent studies have proposed to take advantage of the large sampling of motor unit activity from decomposed HD-EMG signals to extend the synergy concept to the motor neuron level (Madarshahian et al., 2021; Tanzarella et al., 2021; Del Vecchio et al., 2022; Hug et al., 2022; Hug et al., 2023).”*

MINOR COMMENTS

(1) Lines 112-113: Can clarify that hd-EMGs were recorded during isometric conditions at three different ankle angles.

Changed as suggested. The sentence now reads (P.4, L.107): *“For these reasons, we identified and tracked motor neurons innervating the triceps surae muscles (soleus [SOL], gastrocnemius medialis [GM], and gastrocnemius lateralis [GL]) during three isometric contractions performed at different ankle angles.”*

(2) Lines 268-270: Can further clarify how each isolated motor unit was attributed to an individual muscle (GM, GL, or SOL). Was this based solely on the anatomical positions of the channels that contributed to the isolated unit?

Each motor unit was initially associated to the muscle from which the EMG signals were recorded, i.e., based on the anatomical position of the EMG grid. This is a reasonable assumption considering that each grid was positioned over a single muscle. We further confirmed that each motor unit was assigned to the correct muscle by ensuring the absence of crosstalk units, that is, if any unit identified in one muscle actually belonged to the other recorded muscle. When crosstalk was identified (2.6 ± 2.5 MUs per participant), the motor units were discarded from the analysis.

This procedure is described in the manuscript (P.10, L.255):

“We carefully checked all identified motor units for potential crosstalk units or duplicates, thus ensuring the uniqueness of each motor unit within the dataset (Hug et al., 2021b). To this end, we used the discharge times as a trigger to segment and average the HD-EMG signal over a window of 50 ms and identify the motor unit action potentials over the 64 electrodes of each grid. Then, we calculated the average peak-to-peak amplitude of the action potentials for each grid. We considered that a motor unit was falsely associated with a muscle due to crosstalk when the peak-to-peak amplitude was larger in a neighboring muscle than in the muscle of origin. On average, 2.6 ± 2.5 MUs were discarded from the analysis per participant. In addition, we verified that motor units identified from the grid located over the lateral and the medial parts of the SOL were not the same, i.e., duplicates. We considered duplicates as motor units that shared at least 30% of their discharge times with a tolerance of one data point (i.e., 0.5 ms; (Holobar et al., 2014)). At the end of these analyses, motor units identified over the lateral and medial parts of the SOL were pooled, with on average 0.7 ± 1.4 MUs discarded per participant.”

(3) Lines 277-278: Can clarify what exactly was averaged through spike-triggered averaging.

We averaged the 59 single-differential EMG signals from the same grid. The sentence was modified to read (P.10, L.271): *“Thus, we matched motor units using the unique spatiotemporal properties of motor unit action potentials within the grids of electrodes (Martinez-Valdes et al., 2017; Vecchio & Farina, 2019; Oliveira & Negro, 2021). Specifically, EMG signals were first differentiated in the column direction to obtain 59 single-differential signals. The motor unit action potential shapes were identified by spike-triggered averaging each of the 59 EMG signals.”*

(4) Lines 320-321: It is a bit unclear why the criterion stated here is good for finding the 10-s window for cross-correlation.

In order to maximize the statistical power of our analysis, we considered the largest sample of matched motor units across ankle angles. However, some of these motor units sometimes start discharging during the plateau, or have on and off periods that exceed 500ms. Therefore, we first removed from the analysis all the motor units with on/off periods, thus

limiting their impact on the estimation of the level of common input (*Castronovo et al, 2015, J Appl Physiol*). Then, we identified the 10-s window with the highest number of active motor units. Note that we recently showed that a 10-s period is enough to accurately estimate the presence or the absence of a significant level of common input based on cross-correlations (Avrillon et al., 2023, BioRxiv).

Castronovo AM, Negro F, Conforto S & Farina D. (2015). The proportion of common synaptic input to motor neurons increases with an increase in net excitatory input. *J Appl Physiol*. 119, 1337-1346.

Avrillon S, Hug F & Farina D. (2023). A graph-based approach to identify motor neuron synergies. *bioRxiv*, 2023.2002.2007.527433.

(5) Fig. 2 & Fig. 4: Were the signs of any identified PCs flipped according to a certain criterion?

We did not flip the PCs to align them across conditions.

(6) Line 439: It would be informative to report the average and range of the number of motor units identified per subject, for each of the three ankle angles.

This information has been added in the following section (P.17, L.430):

“A total of 595 (37 ± 22 ; range: 5-79), 613 (38 ± 27 ; range: 3-96), and 471 (29 ± 22 ; range: 3-76) motor units were identified at PF20°, PF0°, and DF20°, respectively.”.

Dear Dr Avrillon,

Re: JP-RP-2023-284503R1 "Two motor neuron synergies, invariant across ankle joint angles, activate the triceps surae during plantarflexion" by Jackson Levine, Simon Avrillon, Dario Farina, Francois Hug, and José L. Pons

Thank you for submitting your manuscript to The Journal of Physiology. It has been assessed by a Reviewing Editor and by 2 expert referees and we are pleased to tell you that it is acceptable for publication following satisfactory revision.

REVISION CHECKLIST:

- 'Potential Cover Art' for consideration as the issue's cover image

- Appropriate Supporting Information (Video, audio or data set: see https://jp.msubmit.net/cgi-bin/main.plex?form_type=display_requirements#supp).

We look forward to receiving your revised submission.

Yours sincerely,

Richard Carson
Senior Editor
The Journal of Physiology

REQUIRED ITEMS

- You must start the Methods section with a paragraph headed Ethical Approval. If experiments were conducted on humans confirmation that informed consent was obtained, preferably in writing, that the studies conformed to the standards set by the latest revision of the Declaration of Helsinki, and that the procedures were approved by a properly constituted ethics committee, which should be named, must be included in the article file. If the research study was registered (clause 35 of the Declaration of Helsinki) the registration database should be indicated, otherwise the lack of registration should be noted as an exception (e.g. The study conformed to the standards set by the Declaration of Helsinki, except for registration in a database). For further information see: <https://physoc.onlinelibrary.wiley.com/hub/human-experiments>.

- The Journal of Physiology funds authors of provisionally accepted papers to use the premium BioRender site to create high resolution schematic figures. Follow this link and enter your details and the manuscript number to create and download figures. Upload these as the figure files for your revised submission. If you choose not to take up this offer we require figures to be of similar quality and resolution. If you are opting out of this service to authors, state this in the Comments section on the Detailed Information page of the submission form. The link provided should only be used for the purposes of this submission. Authors will be charged for figures created on this premium BioRender account if they are not related to this manuscript submission.

- Papers must comply with the Statistics Policy: https://jp.msubmit.net/cgi-bin/main.plex?form_type=display_requirements#statistics.

In summary:

- If $n \leq 30$, all data points must be plotted in the figure in a way that reveals their range and distribution. A bar graph with data points overlaid, a box and whisker plot or a violin plot (preferably with data points included) are acceptable formats.

- If $n > 30$, then the entire raw dataset must be made available either as supporting information, or hosted on a not-for-profit repository e.g. FigShare, with access details provided in the manuscript.

- 'n' clearly defined (e.g. x cells from y slices in z animals) in the Methods. Authors should be mindful of pseudoreplication.

- All relevant 'n' values must be clearly stated in the main text, figures and tables, and the Statistical Summary Document (required upon revision).

- The most appropriate summary statistic (e.g. mean or median and standard deviation) must be used. Standard Error of the Mean (SEM) alone is not permitted.

- Exact p values must be stated. Authors must not use 'greater than' or 'less than'. Exact p values must be stated to three significant figures even when 'no statistical significance' is claimed.

- Statistics Summary Document completed appropriately upon revision.

- Please include an Abstract Figure file, as well as the figure legend text within the main article file. The Abstract Figure is a piece of artwork designed to give readers an immediate understanding of the research and should summarise the main conclusions. If possible, the image should be easily 'readable' from left to right or top to bottom. It should show the physiological relevance of the manuscript so readers can assess the importance and content of its findings. Abstract Figures should not merely recapitulate other figures in the manuscript. Please try to keep the diagram as simple as possible and without superfluous information that may distract from the main conclusion(s). Abstract Figures must be provided by authors no later than the revised manuscript stage and should be uploaded as a separate file during online submission labelled as File Type 'Abstract Figure'. Please ensure that you include the figure legend in the main article file. All Abstract Figures should be created using BioRender. Authors should use The Journal's premium BioRender account to export high-resolution

images. Details on how to use and access the premium account are included as part of this email.

EDITOR COMMENTS

Reviewing Editor:

The major comments of both reviewers have been addressed in the revised manuscript. A small number of remaining issues have been identified which should be addressed. These include the interpretation of the new data presented and clarification around the terms 'clusters' and motor neuron synergies.

REFEREE COMMENTS

Referee #1:

The authors have addressed my concerns by removing the clustering analysis and realigning the frameworks of muscle and motor neuron synergies. As they agree that the clustering analysis was not appropriate, I suggest also removing all references to "clusters" in the text (e.g. at lines 62, 95, 102, 649-652) and referring directly to "motor neuron synergies".

Referee #2:

In this revised version, the authors have updated their analysis of the motor neuron synergies with one based on biplots of the motor-unit weights from the principal components (the new Fig. 4) and finding the angles between all pairs of vectors on such biplots (the new Fig. 5). While the authors' effort on addressing the reviewers' comments by implementing such new analyses is much appreciated, some work on suitably interpreting these new results remain to be done.

MAJOR COMMENTS

(1) It is a bit unclear how the new results on Fig. 5 lead to the conclusion that muscles GL and GM "receive the highest proportion of common input" (line 489) (in fact, it is unclear what this very phrase means). If I understand correctly, a consistent small angle between the vectors of the motor-unit PC weights from the same muscle just means that within the same muscle, the proportion of inputs from the two common drives received by the motoneurons tend to be consistent across motor units. (For instance, if motor unit 1 of GL is driven by 4 activation units of drive 1 and 3 units of drive 2, motor unit 2 of GL likewise is driven by a similar proportion of activations from drives 1 and 2.) It is unclear what it means physiologically when the angles from GL and GM units are lower than those from SOL or any of the muscle pairs, and the implications of this finding are not discussed in Discussion section. Does this mean that GL and GM are controlled more like whole-muscles while the SOL motor units are controlled more individually?

(2) Related to the above point, the Discussion section has not been adequately updated in light of the findings of the new analysis. How do the biplot and vector-angle analyses (Fig. 4-5) support the finding that GL-SOL and GM-SOL are the two dominant common inputs? In fact, from the biplots on Fig. 4, it looks like for some subjects and DF angles, GL and GM were also prominently co-represented in a principal component (e.g., in P#5, DF20deg, both GL and GM have units with large PC1 weights). The Discussion section now still reads like one that is based on the PCA/clustering analysis of the previous version. The authors should elaborate on how their new analyses can be related to the conceptual findings.

MINOR COMMENT

In Fig. 4, the vectors denoting motor units from GM and SOL are shown in colors very close to each other (dark blue and light blue, respectively). It is a little difficult to distinguish vectors from the 3 muscles. I suggest using 3 contrasting colors for the three muscles.

END OF COMMENTS

1st Confidential Review

26-Jun-2023

EDITOR COMMENTS

Reviewing Editor:

The major comments of both reviewers have been addressed in the revised manuscript. A small number of remaining issues have been identified which should be addressed. These include the interpretation of the new data presented and clarification around the terms 'clusters' and motor neuron synergies.

We thank the editors and reviewers for their feedback. We have answered their comments below.

REFEREE COMMENTS

Referee #1:

The authors have addressed my concerns by removing the clustering analysis and realigning the frameworks of muscle and motor neuron synergies. As they agree that the clustering analysis was not appropriate, I suggest also removing all references to "clusters" in the text (e.g. at lines 62, 95, 102, 649-652) and referring directly to "motor neuron synergies".

The term 'clusters' has been removed from the manuscript. To clarify the terms used in the manuscript, we consider *motor neuron synergies* as groups of motor units exhibiting covariations in their discharge rates, likely due to their activation by common inputs, previously described as *motor modules*.

Referee #2:

In this revised version, the authors have updated their analysis of the motor neuron synergies with one based on biplots of the motor-unit weights from the principal components (the new Fig. 4) and finding the angles between all pairs of vectors on such biplots (the new Fig. 5). While the authors' effort on addressing the reviewers' comments by implementing such new analyses is much appreciated, some work on suitably interpreting these new results remain to be done.

MAJOR COMMENTS

(1) It is a bit unclear how the new results on Fig. 5 lead to the conclusion that muscles GL and GM "receive the highest proportion of common input" (line 489) (in fact, it is unclear what this very phrase means). If I understand correctly, a consistent small angle between the vectors of the motor-unit PC weights from the same muscle just means that within the same muscle, the proportion of inputs from the two common drives received by the motoneurons tend to be consistent across motor units. (For instance, if motor unit 1 of GL is driven by 4 activation units of drive 1 and 3 units of drive 2, motor unit 2 of GL likewise is driven by a similar proportion of activations from drives 1 and 2.)

It is unclear what it means physiologically when the angles from GL and GM units are lower than those from SOL or any of the muscle pairs, and the implications of this finding are not discussed in Discussion section. Does this mean that GL and GM are controlled more like whole-muscles while the SOL motor units are controlled more individually?

We rephrased the sentence to explain that the smaller the angle formed between two vectors, the higher the proportion of common inputs between the two motor neurons.

P23, L598: *'The covariation of discharge rates between motor units from either GL or GM was particularly high, with a ratio of pairs with significant correlation for all three ankle angles reaching 94.1% and 88.4% for GL and GM, respectively. This is in agreement with the small angles between vectors identified from the PCA (Fig. 5). Together, these observations indicate that the majority of motor neurons innervating either GL or GM received the two main common inputs with a similar weighting.'*

Moreover, the complementary analysis of correlations of discharge rates between pairs of motor neurons showed that only a few pairs of motor units between GL and GM had a consistent significant correlation across conditions. This means that the nervous system may uncouple the behavior of motor neurons from each muscle. We revised the manuscript to better explain how this result might functionally impact the control of forces produced by these muscles.

P23, L603: *'On the contrary, the degree of covariation of discharge rates between motor neurons from GL and GM was lower than for the other pairs of muscles, with 13% of pairs of motor neurons exhibiting significant correlation across all three ankle angles. This is in agreement with previous work that used methods at the muscle level, such as intermuscular coherence (Hug et al., 2021b; Rossato et al., 2022) or PCA (Mazzo et al., 2022), to assess the covariations of motor unit discharge rates in the triceps surae muscles. Similarly, pairs of motor units from the SOL appear to have lower covariations of discharge rates than from GL and GM. The nervous system may thus take advantage of the geometry and compartmentalization of this muscle (Bolsterlee et al., 2018) to control multiple sources of common input and produce forces in multiple directions (Segal et al., 1991; English et al., 1993). Such strategies have been observed in muscles controlling the individual fingers through multiple tendons (Keen & Fuglevand, 2004; Hockensmith et al., 2005; McIsaac & Fuglevand, 2007), or in the biceps brachii producing force for forearm supination and/or elbow flexion (Barry et al., 2009).'*

(2) Related to the above point, the Discussion section has not been adequately updated in light of the findings of the new analysis. How do the biplot and vector-angle analyses (Fig. 4-5) support the finding that GL-SOL and GM-SOL are the two dominant common inputs? In fact, from the biplots on Fig. 4, it looks like for some subjects and DF angles, GL and GM were also prominently co-represented in a principal component (e.g., in P#5, DF20deg, both GL and GM have units with large PC1 weights). The Discussion section now still reads like one that is based on the PCA/clustering analysis of the previous version. The authors should elaborate on how their new analyses can be related to the conceptual findings.

We now have updated the discussion to better link our results to the concepts of motor modules and muscle/motor neuron synergies. Specifically, i) we removed all the sentences arguing that the two synergies are grouping motor neurons from the GL/SOL and GM/SOL, and ii) we linked the invariance of motor neuron synergies across conditions to the existence of motor modules as building block for movement control.

P24, L634: *'Here, PCA identified two principal components that explained the main variations of discharge rates of the identified motor neurons. It is worth noting that Gibbs et al. (1995) have already reported significant levels of synchronization between the discharge*

times of motor units from the gastrocnemius and the soleus muscles, consistent with our results outlining the projection of common inputs to the two muscles.

The results of this study support three of the hypotheses proposed by Hug et al. (2023) within the motor neuron synergies framework, namely that during movement 1) motor neurons receives common input in relatively large groups; 2) motor neuron synergies may significantly differ from the classical definition of motor neuron pools, such that synergies may span across muscles (i.e., GL and SOL, GM and SOL) and/or involve only a portion of a muscle (i.e., SOL); and 3) synergies represent motor modules used by the central nervous system to reduce the dimensionality of control. The latter observation is the main novelty of our study. Previous studies identified motor neuron synergies during isometric contractions performed at a single joint angle (Madarshahian et al., 2021; Del Vecchio et al., 2022; Hug et al., 2022) and, therefore, did not demonstrate their invariance across tasks, which is needed to prove a modular organization of movement control (d'Avella et al., 2003; d'Avella & Bizzi, 2005).'

MINOR COMMENT

In Fig. 4, the vectors denoting motor units from GM and SOL are shown in colors very close to each other (dark blue and light blue, respectively). It is a little difficult to distinguish vectors from the 3 muscles. I suggest using 3 contrasting colors for the three muscles.

The figure has been updated to increase the contrast between the colors of each muscle.

Dear Dr Avrillon,

Re: JP-RP-2023-284503R2 "Two motor neuron synergies, invariant across ankle joint angles, activate the triceps surae during plantarflexion" by Jackson Levine, Simon Avrillon, Dario Farina, Francois Hug, and José L. Pons

We are pleased to tell you that your paper has been accepted for publication in The Journal of Physiology.

Authors should note that it is too late at this point to offer corrections prior to proofing. The accepted version will be published online, ahead of the copy edited and typeset version being made available. Major corrections at proof stage, such as changes to figures, will be referred to the Editors for approval before they can be incorporated. Only minor changes, such as to style and consistency, should be made at proof stage. Changes that need to be made after proof stage will usually require a formal correction notice.

Yours sincerely,

Richard Carson
Senior Editor
The Journal of Physiology

P.S. - You can help your research get the attention it deserves! Check out Wiley's free Promotion Guide for best-practice recommendations for promoting your work at www.wileyauthors.com/eeo/guide. You can learn more about Wiley Editing Services which offers professional video, design, and writing services to create shareable video abstracts, infographics, conference posters, lay summaries, and research news stories for your research at www.wileyauthors.com/eeo/promotion.

IMPORTANT NOTICE ABOUT OPEN ACCESS: To assist authors whose funding agencies mandate public access to published research findings sooner than 12 months after publication, The Journal of Physiology allows authors to pay an Open Access (OA) fee to have their papers made freely available immediately on publication.

You can check if your funder or institution has a Wiley Open Access Account here: <https://authorservices.wiley.com/author-resources/Journal-Authors/licensing-and-open-access/open-access/author-compliance-tool.html>.

EDITOR COMMENTS

Reviewing Editor:

All comments have been satisfactorily addressed.

REFeree COMMENTS

Referee #1:

The authors have fully addressed my comments.

Referee #2:

The authors have addressed all of my concerns over two rounds of revision. I have no further comments.

2nd Confidential Review

03-Aug-2023